# Fibrinogen function achieved through multiple covalent states

Diego Butera[1] & Philip J. Hogg [1,2] ✉

Disulfide bonds link pairs of cysteine amino acids and their formation is assumed to be complete in the mature, functional protein. Here, we test this assumption by quantifying the redox state of disulfide bonds in the blood clotting protein fibrinogen. The disulfide status of fibrinogen from healthy human donor plasma and cultured human hepatocytes are measured using differential cysteine alkylation and mass spectrometry. This analysis identifies 13 disulfide bonds that are 10–50% reduced, indicating that fibrinogen is produced in multiple disulfide-bonded or covalent states. We further show that disulfides form upon fibrin polymerization and are required for a robust fibrin matrix that withstands the mechanical forces of flowing blood and resists premature fibrinolysis. The covalent states of fibrinogen are changed by fluid shear forces ex vivo and in vivo, indicating that the different states are dynamic. These findings demonstrate that fibrinogen exists and functions as multiple covalent forms.

[1] The Centenary Institute, Camperdown, NSW 2050, Australia. [2] NHMRC Clinical Trials Centre, University of Sydney, Sydney, NSW 2006, Australia.
✉email: phil.hogg@sydney.edu.au

Protein Data Bank structures currently contain >180,000 disulfide bonds[1]. In human proteins, these bonds are found mostly in membrane proteins (plasma and organelle) and proteins that are secreted by cells. Although, a growing number of disulfide bonds are being identified in proteins that function in the nucleus or cytoplasm. Acquisition of disulfide bonds contributed significantly to the evolution of vertebrates[2]. Most disulfides in human proteins were acquired in vertebrate ancestors and very often coincided with formation of a new protein. These bonds were almost never lost as the protein evolved and are continuing to be acquired today[3,4].

Disulfide bonds form during maturation of proteins in the cell and influence protein folding in ways that are not completely understood. Three different roles for these bonds in protein folding have been proposed: one where disulfides drive global protein folding[5], another where these bonds direct protein folding in a stepwise fashion[6–8], and a third where disulfide bond formations stabilize the native conformation rather than facilitate folding[9–11]. Thousands of disulfide-bonded protein crystal structures have revealed that these bonds are almost invariably intact.

Our laboratory has focused on the identification and study of allosteric disulfides, which are a subset of disulfide bonds that regulate mature protein function when cleaved or formed[1]. The study of these functional bonds has been facilitated by the development of a differential cysteine alkylation and mass spectrometry method to precisely quantify the redox state of disulfides in proteins in their native environments[12]. In the course of our studies, we were identifying an increasing number of labile disulfide bonds in proteins that from structural studies have been assumed to be intact in the mature protein. These unexpected findings have been systematically investigated in this study.

In this work, we examine the blood protein fibrinogen, which plays a critical role in arresting bleeding (hemostasis) and obstruction of blood vessels (thrombosis) by bridging platelets at sites of vessel injury. We report that fibrinogen exists in multiple disulfide-bonded states in the circulation that are important for the function of the protein.

## Results

**Fibrinogen exists in multiple disulfide-bonded states in the circulation.** Fibrinogen was selected for study because of its relative abundance in blood and its well defined quaternary structure[13] that contains 17 disulfide bonds. The protein consists of three pairs of polypeptide chains; two Aα, two Bβ and two γ chains. Two AαBβγ units are linked head-to-head to form a structure with a central E region and two peripheral D regions. The disulfide bonds in the protein are located in the globular E (7 bonds) and D regions (10 bonds).

The redox state of cysteine thiols in fibrinogen were frozen before the protein was removed from its ex vivo plasma environment. Blood from healthy donors was drawn by venipuncture into citrate as anti-coagulant, plasma prepared by centrifugation and fibrinogen collected on antibody-coated magnetic beads (Fig. 1a). The unpaired cysteine thiols in bead-bound fibrinogen were alkylated with 2-iodo-N-phenylacetamide ($^{12}$C-IPA), the protein isolated by sodium dodecyl sulfate polyacrylamide gel electrophoresis (SDS-PAGE; Fig. 1b), the disulfide bonds reduced with dithiothreitol, and the disulfide-bonded cysteine thiols alkylated with a stable carbon-13 isotope of IPA ($^{13}$C-IPA)[14]. We attempted to alkylate fibrinogen in blood at the point of collection, however, the $^{12}$C-IPA solvent, dimethyl sulfoxide (DMSO), caused hemolysis of red blood cells[15].

The fibrinogen was digested with trypsin and peptides quantified by HPLC and identity established by mass spectrometry (Supplementary Fig. 1). The levels of the different redox forms of the cysteines was determined from the relative abundance of peptides labeled with $^{12}$C-IPA and/or $^{13}$C-IPA (Supplementary Table 1 and Supplementary Data 1).

The redox state of 14 of the 17 fibrinogen disulfides was quantified: 5 in the E region and 9 bonds in the D region (Fig. 1c). Of the 14 bonds that were measured, 13 have been structurally defined[13]. The D region αC491 is predicted to pair with αC461, although this bond has not been structurally characterized and has been omitted from this analysis. Surprisingly, all 13 disulfide bonds exist in bound (oxidized) or cleaved (reduced) forms in the fibrinogen populations of ten healthy human donors (4 male, 6 female, 22–58 years old) (Fig. 1d). The bonds ranged from 10 to 50% reduced and there was remarkably small donor-to-donor variation and no significant gender difference. The coefficients of variation ranged from a low of 3.9% for the βC424-βC437 disulfide to a high of 16.5% for the αC68-βC106 bond.

Our analysis of disulfide redox states is reliant on the disulfide pairing defined from crystal structures of fibrinogen. We sought confirmation of the disulfide pairing in the soluble protein from mass spectrometry analysis of disulfide-linked peptides. Fibrinogen was immunoprecipitated from a healthy 25-year-old male donor plasma, digested with trypsin and peptides resolved by HPLC and analyzed by mass spectrometry. Thirteen disulfide-linked peptides encompassing 7 of the 14 disulfides measured in the redox state analysis were identified (Supplementary Table 3), which is in accordance with the disulfide pairing observed in crystal structures. We next examined whether fibrinogen arrives in blood as multiple covalent states, or a single covalent form is secreted by liver hepatocytes, the main source of circulating fibrinogen, and then changed in the circulation.

**Fibrinogen secreted by cultured hepatocytes exists in multiple disulfide-bonded states.** The redox state of fibrinogen constitutively produced by the HepG2 human hepatocyte cell line was analyzed. Serum-free conditioned medium was collected for 18 h under either standard normoxic (18.8% $O_2$) or hypoxic (1% $O_2$) conditions. Hypoxic conditions were employed to test for possible $O_2$-mediated oxidation of disulfide bonds in the secreted fibrinogen. The fibrinogen in the conditioned medium was collected on antibody-coated magnetic beads and analyzed as for the plasma protein (Fig. 1a). The disulfide bond status of the cell-derived fibrinogen is indistinguishable from plasma fibrinogen produced under either normoxic or hypoxic conditions (Supplementary Fig. 2). This finding indicates that fibrinogen arrives in plasma in defined covalent states and is not a consequence of post-secretion redox events in the blood.

These results imply that circulating fibrinogen exists in multiple disulfide-bonded states. A protein containing $n$ disulfide bonds, where the bonds are either formed or broken, has $2^n$ possible disulfide-bonded states. This situation is illustrated in a model polypeptide containing 5 disulfide bonds that can exist in 32 ($2^5$) possible disulfide-bonded states (Supplementary Fig. 3a). The different states are represented in cartoon form in Supplementary Fig. 3b. In the case of the 13 fibrinogen disulfides that were measured, this analysis equates to a maximum 8192 possible disulfide-bonded states of the protein. The functional relevance of this biology was examined by measuring changes in the covalent states when fibrinogen is converted to fibrin polymer.

**Disulfide bonds form during fibrin polymer formation.** Fibrin forms the scaffold of clots. Formation of fibrin is triggered by thrombin cleavage of fibrinopeptides A and B from the N-termini of the Aα and Bβ chains of fibrinogen (Fig. 2a). Removal of the fibrinopeptides exposes "knobs" that bind to "holes" in the γ- and β-nodules to initiate fibrin fiber formation (Fig. 2a).

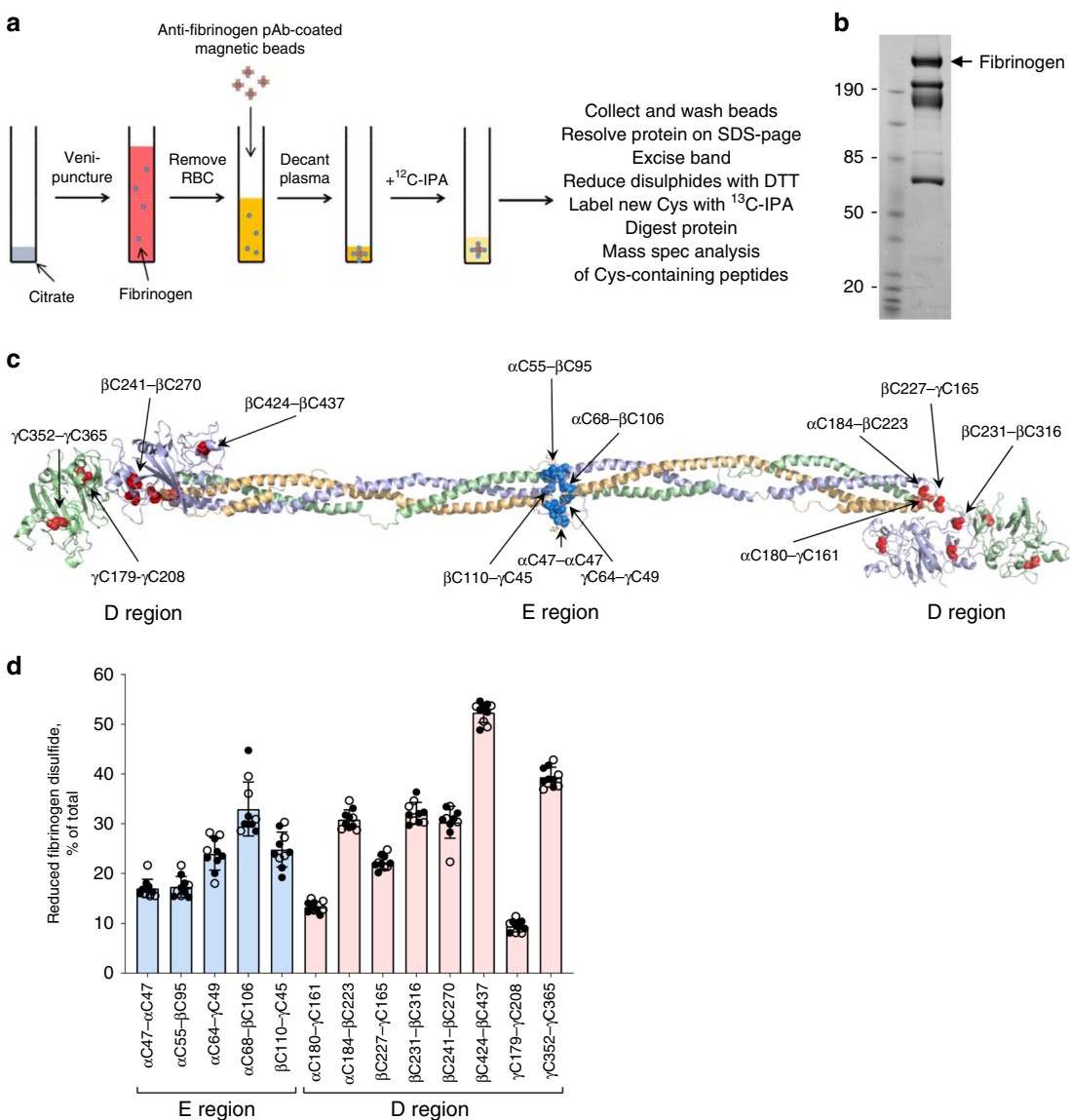

**Fig. 1 Fibrinogen exists in multiple disulfide-bonded states in the circulation. a** Blood from healthy donors was drawn by venipuncture into citrate as anti-coagulant, plasma prepared by centrifugation and fibrinogen collected on antibody-coated magnetic beads. The unpaired cysteine thiols in bead-bound fibrinogen was alkylated with $^{12}$C-IPA, the protein resolved on SDS-PAGE and the disulfide-bonded cysteine thiols alkylated with $^{13}$C-IPA following reduction with DTT. The protein was digested with trypsin, 17 peptides (Table S1) encompassing cysteines representing 13 of the 17 fibrinogen disulfide bonds were analyzed by HPLC and mass spectrometry and the redox state of the disulfides quantified. **b** Example of $^{12}$C-IPA-labeled fibrinogen resolved on SDS-PAGE. Molecular mass standards are shown in the left-hand lane. Fibrinogen has a molecular mass of 340 kDa. **c** Ribbon structure of fibrinogen and the positions of the 13 disulfide bonds (red spheres for the D region disulfides and cyan spheres for the E region bonds) that were mapped[13] (PDB identifier 3ghg). The α-chains are in wheat, β-chains in light blue and γ chains in light green. **d** Redox states of the five E region and eight D region disulfides in ten healthy human donors (six female—solid symbols; four male—open symbols). The bars and errors are mean ± SD. Source data are provided as a Source Data file.

Fibrin was produced in four of the ten healthy human donor plasmas (Fig. 1d) by adding thrombin. The fibrin polymer was collected on a plastic rod, cysteines immediately alkylated with $^{12}$C-IPA and the redox state of the fibrin disulfides quantified and compared to the redox state of the parent fibrinogen. All 13 disulfides are significantly more oxidized in fibrin polymer (Fig. 2b). We considered two scenarios that could account for this change: (1) a subset of fibrinogen covalent states containing more oxidized disulfide bonds is preferentially used for fibrin formation, or (2) disulfide bonds form during fibrin formation. The first scenario implies there would be enrichment of reduced fibrinogen states as fibrin polymer forms in plasma. This was tested by adding a limiting concentration of thrombin to healthy

donor plasma, the fibrin polymer removed, another aliquot of thrombin added to the depleted plasma and the new fibrin polymer removed. Fibrinogen concentration in the plasma was depleted 41% in the first reaction and >95% in the second reaction. There was no change in the disulfide-bonded states of fibrinogen as fibrin formed (Fig. 3a), although the fibrin polymer produced is more oxidized. This result supports the second scenario that disulfide bonds oxidize during fibrin formation. We next determined whether disulfides form before or after fibrin polymerizes.

Thrombin was added to healthy donor plasmas and the resulting fibrin was prevented from polymerizing by the tetrapeptide inhibitor, GPRP. GPRP binds to the γ-nodule "hole" and blocks

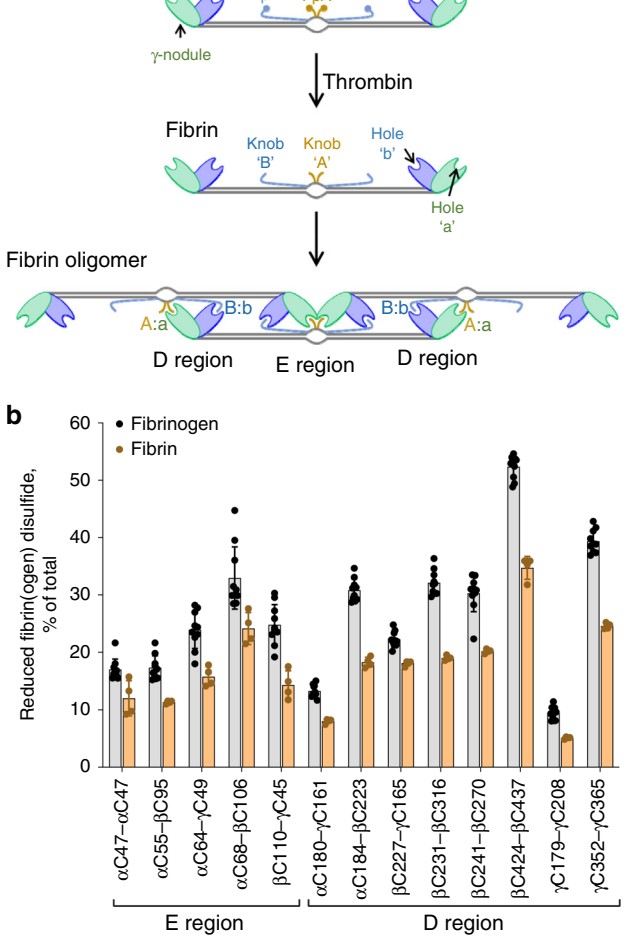

**Fig. 2 The disulfides in fibrin polymer are more oxidized than the bonds in fibrinogen. a** Conversion of fibrinogen to fibrin is mediated by thrombin proteolysis of fibrinopeptides (Fp) A and B in the central E region to form knobs "A" and "B". Fibrin oligomers are formed when the exposed knobs "A" and "B" bind to holes "a" and "b" in the γ- and β-nodules, respectively. **b** Redox states of the five E region and eight D region disulfides in fibrinogen (10 donors) and fibrin polymer (in 4 of the 10 donors). The bars and errors are mean ± SD. All 13 disulfides are more oxidized in fibrin polymer: $p < 0.05$ for the αC68-βC106 bond and $p < 0.01$ for the other 12 bonds. Parametric unpaired $t$-test was used to evaluate differences between groups. Source data are provided as a Source Data file.

fibrin binding. There is no oxidation of disulfide bonds in the fibrin if polymerization is prevented (Fig. 3b), indicating that disulfides form as a result of fibrin polymerization. We next investigated the functional consequences of disulfide formation during fibrin polymer formation.

**Preventing disulfide bond formation alters structure and function of the fibrin polymer.** As functional studies of fibrinogen require the isolated protein, we first examined the effect of removing fibrinogen from plasma on the redox state of the fibrinogen disulfides. Two preparations of purified fibrinogen were tested, one that was stored lyophilized and another that was stored frozen at −20 °C. All 13 disulfide bonds of the purified fibrinogens were more oxidized ($p < 0.01$) than the bonds in healthy donor fibrinogens prepared directly from plasma (Supplementary Fig. 4). This result suggests that removal of fibrinogen from plasma, an

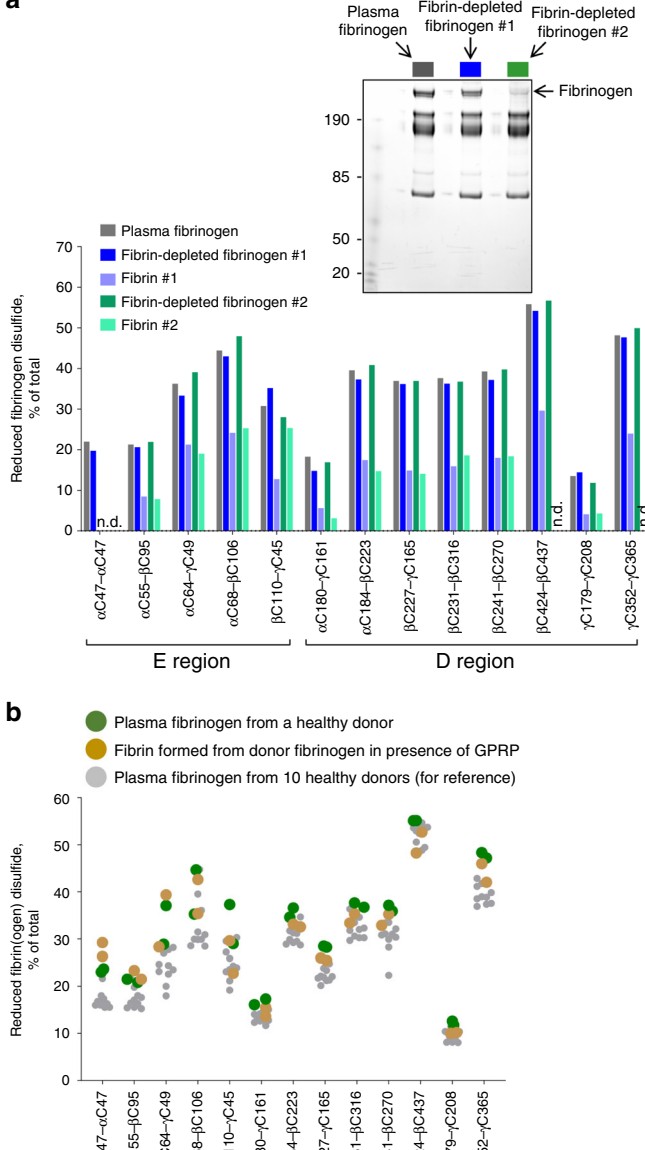

**Fig. 3 Fibrinogen disulfides form upon fibrin polymer formation. a** All covalent states of fibrinogen contribute to fibrin formation. An aliquot of thrombin was added to healthy donor plasma and the fibrin polymer removed. Another aliquot of thrombin was added to the depleted plasma and the new fibrin polymer removed. At top is a Coomassie stained gel showing successive depletion of fibrinogen from the plasma. Molecular mass standards are shown in the left-hand lane. Fibrinogen concentration in the plasma was reduced 41% in the first reaction and >95% in the second reaction. There was no change in the disulfide-bonded states of fibrinogen as fibrin polymer was formed, whereas the disulfides in fibrin polymer were more oxidized. Peptides that were not resolved in the experiment are indicated as not determined (n.d.). Source data are provided as a Source Data file. **b** Fibrin polymerization is required for disulfide bonds to form. Thrombin was added to a male and a female healthy donor plasma (green symbols) and the resulting fibrin (brown symbols) was prevented from polymerizing by the tetrapeptide, GPRP. There is no oxidation of disulfide bonds in the fibrin if polymerization is prevented. The redox states of ten healthy donor fibrinogens are shown for reference (gray symbols). Source data are provided as a Source Data file.

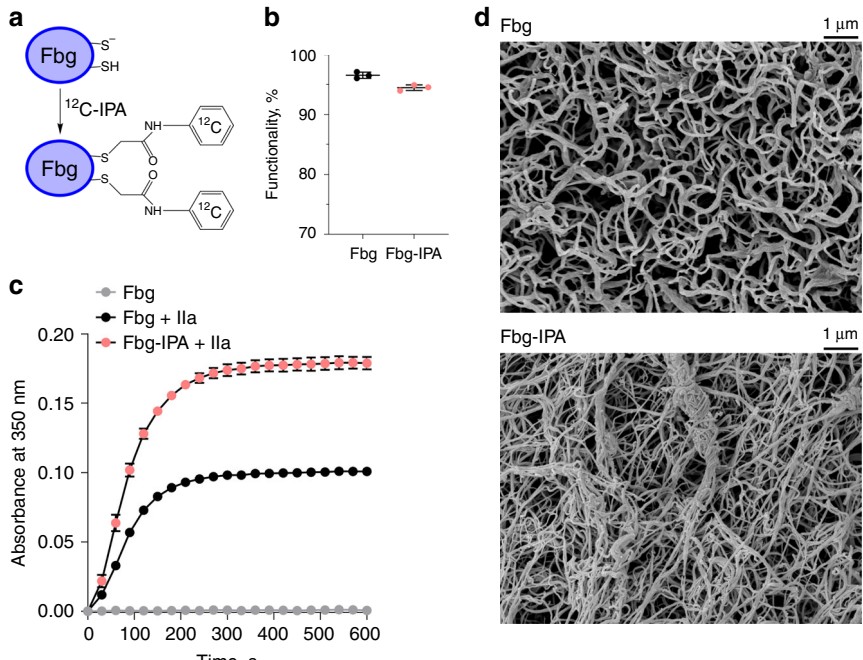

**Fig. 4 Prevention of disulfide bond formation during fibrin formation alters the polymer structure. a** Purified fibrinogen (Fbg) was treated with $^{12}$C-IPA (Fbg-IPA) to block the unpaired cysteine thiols in the protein. Labeling conditions were identical to that for the mass spectrometry experiments (Figs. 1–5). **b** Alkylation of the unpaired cysteine thiols in fibrinogen with IPA does not impair the fibrin forming capacity of the protein. The data points are the mean ± SD from three independent experiments and represent the percentage of Fbg/Fbg-IPA consumed in the formation of fibrin polymer. Source data are provided as a Source Data file. **c** Kinetics of fibrin polymerization from Fbg and Fbg-IPA measured by light scattering. The data points are the mean ± SD from three independent experiments. Source data are provided as a Source Data file. **d** Representative structures from $n = 20$ images of fibrin matrices from Fbg and Fbg-IPA characterized by scanning electron microscopy.

environment with a high-redox buffering capacity, can result in air oxidation of the protein. The lyophilized protein was used for the functional studies.

The unpaired cysteine thiols in purified fibrinogen were alkylated with $^{12}$C-IPA to prevent disulfide bond formation during fibrin polymer formation (Fig. 4a). IPA alkylation of fibrinogen did not impair the fibrin forming capacity of the protein (Fig. 4b): 96.6 ± 0.5% (SD) and 94.5 ± 0.5% of the fibrinogen and fibrinogen-IPA, respectively, was consumed in formation of fibrin polymer. The kinetics of fibrin polymerization measured by light scattering is clearly different for fibrinogen-IPA (Fig. 4c). The time to reach half-maximal turbidity was the same for fibrinogen and fibrinogen-IPA (80 s) whereas that maximal extent of turbidity was 1.8-fold higher for fibrinogen-IPA. To determine the structural basis which underlies this difference, the fibrin matrices were examined by scanning electron microscopy (Fig. 4d). The architecture of the fibrin-IPA matrix is obviously different from control fibrin. The individual fibrin-IPA polymer strands have a higher density, are thinner (0.07 ± 0.01 μm, $n = 20$) than control fibrin (0.11 ± 0.02 μm, $n = 20$) and less organized. In addition, fibrin-IPA strands can form large rope-link bundles consisting of 4 to >10 fibrin strands. These large bundles likely account for the higher light scattering of the fibrin-IPA polymer. The differences in these fibrin structures is reflected in their differences in permeability, compaction and susceptibility to plasmin lysis.

Fibrin-IPA matrices are more permeable than control fibrin matrices, with Darcy constants of $2.81 ± 0.74 \times 10^{-13}$ cm$^2$ and $6.84 ± 1.26 \times 10^{-13}$ cm$^2$ for fibrin and fibrin-IPA, respectively (Supplementary Fig. 5a). Fibrin-IPA matrices are also more easily compacted by centripetal forces than control fibrin (Supplementary Fig. 5b) and more readily lysed by plasmin, with fibrinolysis clearing zones of 33.7 ± 7.2 mm$^2$ and 50.1 ± 3.2 mm$^2$ for fibrin

and fibrin-IPA, respectively (Supplementary Fig. 5C). An important question is whether the different fibrinogen states are static or dynamic, that is can they change in their plasma environment. Fibrinogen is subjected to mechanical shear forces in the circulation, so we tested the effect of this external stimulus on the covalent states of the protein.

**Fluid shear forces change the covalent states of fibrinogen ex vivo and in patients with heart failure.** Healthy donor plasma was subjected to fluid shear forces found in arterioles (2000 s$^{-1}$) and stenotic blood vessels (10000 s$^{-1}$)[16]. The 2000 s$^{-1}$ shear rate did not change the redox state of the 13 fibrinogen disulfides, while the pathological fluid shear caused significant reduction of all 13 disulfides ($p < 0.01$) (Fig. 5a). The high-fluid shear, therefore, triggered global cleavage of fibrinogen disulfides, indicating that the bonds are dynamic in plasma. We next sought evidence for this change in humans in vivo.

A clinical situation associated with high-blood shear is the treatment of refractory heart failure patients with extracorporeal membrane oxygenation (ECMO) support. The ECMO device subjects blood to high shear and both thrombosis and bleeding can occur with this intervention[17]. Eight of the 13 fibrinogen disulfide bonds were significantly ($p < 0.001$) more reduced in patients receiving ECMO than in healthy human donors (Fig. 5b). These differences are notable considering the variability in disease severity and therapeutic intervention in these patients and suggest that reduction of the fibrinogen disulfides is a result of the blood shear associated with the device. Importantly, this finding indicates that the covalent forms of fibrinogen can change in the circulation, which has implications for hemostatic disorders in general. A key consideration is whether the multiple disulfide-bonded states of fibrinogen are a feature of other circulating proteins.

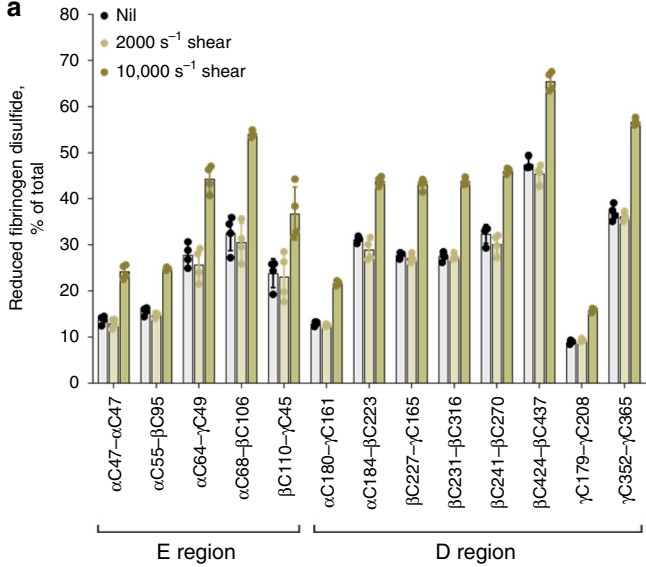

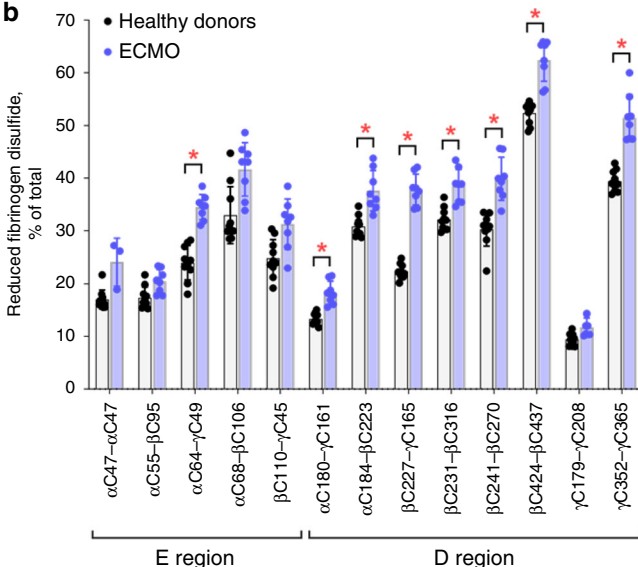

**Fig. 5 Fibrinogen covalent states are changed by mechanical shearing in vitro and in heart failure patients subjected to ECMO. a** Redox states of the five E region and eight D region fibrinogen disulfides of four different healthy donor plasmas sheared at rates of $2000\ s^{-1}$ or $10000\ s^{-1}$ for 5 min. The bars and errors are mean ± SD. Shearing at $10000\ s^{-1}$ results in reduction of all 13 fibrinogen disulfides ($p < 0.01$ for the βC110-γC45 bond and $p < 0.001$ for the other disulfides). Parametric unpaired $t$-test was used to evaluate differences between groups. Source data are provided as a Source Data file. **b** Redox states of the five E region and eight D region fibrinogen disulfides in healthy donors ($n = 10$) versus patients with severe congestive cardiac failure subjected to ECMO ($n = 8$). The bars and errors are mean ± SD. Significant differences ($p < 0.001$) in the redox state of fibrinogen disulfides are indicated by stars. Parametric unpaired $t$-test was used to evaluate differences between groups. Source data are provided as a Source Data file.

**α2-Macroglobulin also exists in multiple covalent states in the circulation.** α2-macroglobulin is a broad-spectrum endopeptidase inhibitor[18] produced by the liver and macrophages and circulates in blood at a concentration of ~2.7 mg/mL. It regulates proteolysis in several biological processes, including nutrition, tissue remodeling and infection. As we observed for fibrinogen, α2-macroglobulin also exists as multiple disulfide-bonded states

in the circulation. The enzyme contains 12 structurally defined disulfide bonds[19] and we were able to quantify the redox state of all 12. The 12 bonds range from 10 to 70% reduced in eight healthy human donors (three male, five female, 18–48 years old) (Supplementary Table 4, Supplementary Data 2 and Fig. 6). As observed for the fibrinogen disulfides, there is very small donor-to-donor variation and no significant gender difference in the redox states of the α2-macroglobulin disulfides. The coefficients of variation ranged from a low of 2.5% for the C642-C689 disulfide to a high of 10.8% for the C847-C883 bond.

## Discussion

Our findings demonstrate that the polypeptide backbone of fibrinogen exists not in a single covalent form in the circulation but rather many, possibly hundreds, of different covalent forms based on whether individual disulfide bonds are formed or not. A striking feature of the different covalent states of fibrinogen is how little the balance of the states differ from donor to donor. The coefficients of variation ranged from 3.9% to 16.5% for 13 fibrinogen disulfide bonds in 10 healthy donors and is not appreciably influenced by donor age or gender. Compelling questions are how and why has fibrinogen evolved to exist in multiple disulfide-bonded states?

One possibility is to facilitate folding and maturation. Protein folding is a spontaneous process that is dominated by the second law, $\Delta G = \Delta H - T\Delta S$, where a net negative Gibbs free energy (G) is achieved through weighing of the enthalpy ($\Delta H$) and entropy ($-T\Delta S$) contributions. The conformational entropies of the more reduced disulfide-bonded states of fibrinogen are predicted to be higher than the more oxidized states. This may mitigate to some extent the negative entropic effects of folding. It is possible that these entropic gains were a driving force for the evolution of these protein states. If the balance of the different covalent forms are dictated by thermodynamic considerations, this could account for the small inter-individual variation.

Fibrinogen is the precursor for fibrin polymer that reinforces thrombi against the fluid shear forces of the circulation. We observed that disulfide bonds in fibrin polymer are generally more oxidized than in the parent fibrinogen. It was possible that a subset of fibrinogen covalent states containing more oxidized disulfide bonds was preferentially used for fibrin formation, or that disulfide bonds form during fibrin formation. We found no preference for more oxidized forms of fibrinogen for fibrin polymer formation and there is no oxidation of disulfide bonds if soluble fibrin is prevented from polymerizing. These findings imply that disulfide bonds form as fibrin polymerizes.

To examine the importance of disulfide formation for fibrin structure and function, the unpaired cysteine thiols in fibrinogen were alkylated with IPA to block bond formation during fibrin formation. The fibrin matrix formed from the alkylated fibrinogen had markedly different structure, was leaky, fragile and more susceptible to proteolysis. These results suggest that disulfide bond formation during fibrin polymerization is necessary to produce a robust fibrin matrix that can withstand the mechanical forces of flowing blood and resist premature fibrinolysis. We cannot exclude the possibility, however, that IPA alkylation of one or more cysteines results in steric effects on fibrin polymer formation.

Formation of disulfides upon fibrin polymerization may relate to requirement for more rigid D region nodule holes. For instance, the γC352-γC365 and βC424-βC437 disulfide bonds line the γ- and β-nodule knob-binding pockets, respectively, and adjacent calcium-binding pockets (Supplementary Fig. 6). Calcium binding promotes lateral aggregation of fibrin polymers[20] and reinforces the γ-nodule hole against mechanical forces[21].

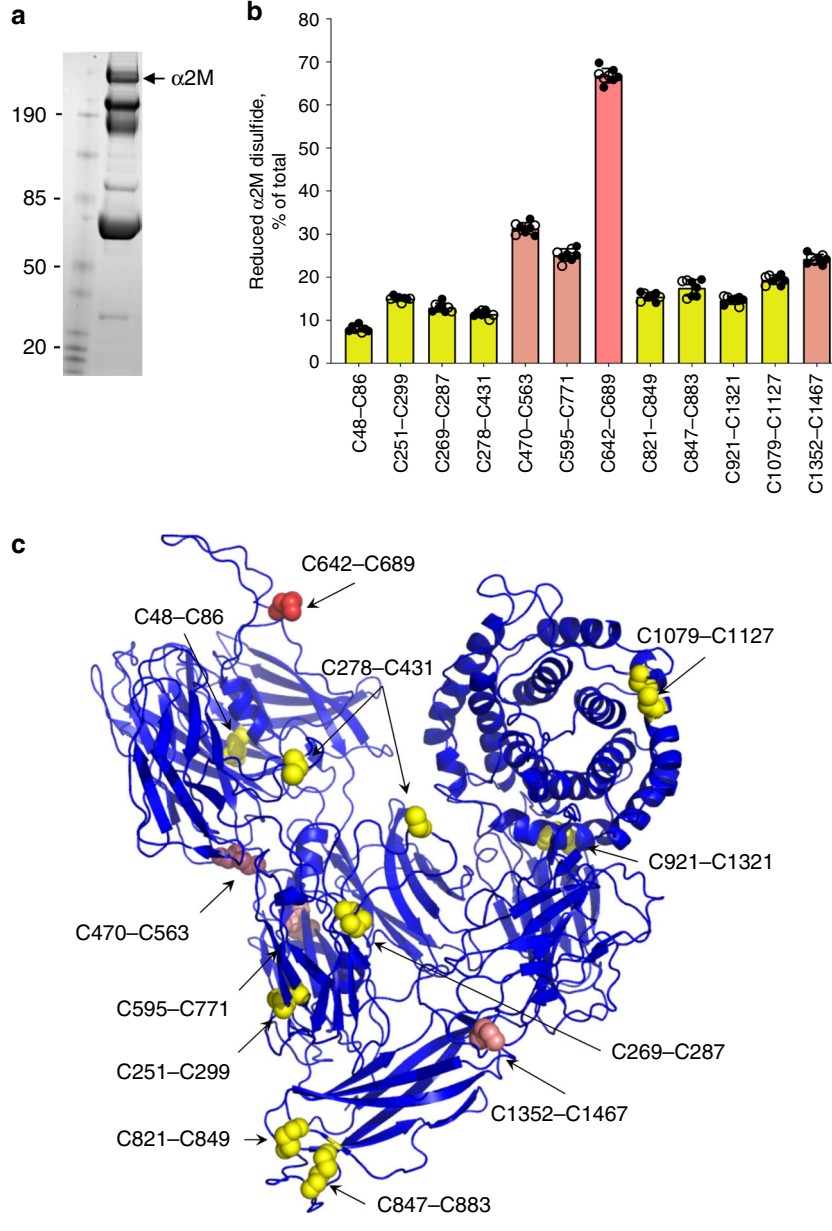

**Fig. 6 α2-Macroglobulin also exists in multiple disulfide-bonded states in the circulation. a** α2-Macroglobulin was isolated and processed from the same healthy donor plasmas as described for fibrinogen in Fig. 1. Shown is a representative [12]C-IPA-labeled α2-macroglobulin resolved on SDS-PAGE. Molecular mass standards are shown in the left-hand lane. α2-Macroglobulin is a homo-tetramer with a molecular mass of 720 kDa. **b** Redox states of the 12 α2-macroglobulin disulfide bonds in eight healthy human donors (five female—solid symbols; three male—open symbols). The bars and errors are mean ± SD. Bonds indicated by yellow bars are <20% reduced, salmon bars 20–40% reduced and the red bar >40% reduced. Source data are provided as a Source Data file. **c** Ribbon structure of α2-macroglobulin and the positions of the 12 disulfide bonds (spheres)[19] (PDB identifier 4acq). The disulfide colour matches the coding in part B. The C278-C431 disulfide is cleaved in this structure, while the other 11 disulfide bonds are intact.

Rigid disulfide-bonded binding pockets may be required to withstand the mechanical stresses associated with formation of a fibrin mesh[22]. These findings suggest the multiple covalent forms of fibrin and the inherent conformational flexibility therein enable efficient interactions and alignments of fibrin molecules, and the subsequent disulfide bond formation results in a productive fibrin polymer matrix.

Our finding of multiple disulfide-bonded states of fibrinogen is distinct from the contribution of allosteric disulfides to protein function. Allosteric disulfides are a subset of disulfide bonds that control mature protein function when cleaved or formed[1,23]. About 40 allosteric disulfides have been identified to date across biological systems and lifeforms[1]. These bonds, which typically

are just one of the disulfide bonds in a protein[12,14], are nearly always on the protein's surface. This position enables their redox state to be controlled by protein oxidoreductases[24,25]. Twelve of the 13 fibrinogen disulfide bonds, however, are mostly hidden from solvent in the interior of the structure (Supplementary Fig. 7). The exception is the E domain α47-α47 disulfide. This indicates that majority of the bonds in the mature protein will be inaccessible to the oxidoreductases and implies their redox state is set during folding and maturation. The observation that fibrinogen secreted by cultured hepatocytes has the same disulfide status as the plasma protein supports this conclusion.

Importantly, the different covalent forms of fibrinogen can change in response to fluid shear forces experienced in the

circulation. The fluid shear results in global reduction of fibrinogen disulfide bonds. Cleavage of disulfide bonds is a highly directional chemistry that is influenced by distortion of the protein in which the bond resides[12,26,27]. Mechanical strain changes the alignment of the sulfur atoms involved in the cleavage and alters the rate of bond cleavage[28,29]. In the case of fibrinogen, mechanical shearing promotes bond cleavage. Future studies will explore the role of accessory plasma factors that are involved in the disulfide cleavage. Notably, perturbation of the mix of the covalent forms of fibrinogen could tip the balance between bleeding and clotting in individuals and underlie some hereditary or acquired thrombophilia's[30].

Fibrinogen is not alone in existing in multiple covalent states as the circulating endopeptidase inhibitor, α2-macroglobulin, also exists in multiple disulfide-bonded states. It is very likely that this biology will be found in many soluble and intrinsic membrane proteins. This will have broad implications for drug resistance and drug development. If a drug binds preferentially to one or more covalent forms of a protein, then events that reduce the incidence of these forms in individuals would result in drug resistance. Similarly, if drugs are developed against some covalent forms of a protein target but not others, this will inevitably result in drug resistance in some individuals. In experimental settings, the multiple covalent states of a protein should be considered when these proteins are investigated. Antibodies and small molecule inhibitors may efficiently bind and inhibit some of the covalent states but not others.

It was recognized 75 years ago by Erin Schrödinger that proteins have enough potential variety in their configurations to encode huge amounts of information[31]. Existing in multiple disulfide-bonded states is an effective and efficient way of maximizing the variety of information a single protein can convey.

## Methods

**Blood collection and processing.** All procedures involving collection of human blood from healthy volunteers were in accordance with the Human Research Ethics Committee of the University of Sydney (approval HREC 2014/244) and informed consent was obtained from all individuals. All procedures involving collection of human blood from ECMO patients were in accordance with the Alfred Hospital Ethics, Monash University Standing Committee for Research in Humans (approval 388/13) and informed consent was obtained from all individuals. All procedures were in accordance with the Helsinki Declaration of 1983. Blood was collected by venesection using a 21 g butterfly Terumo needle from 10 healthy donors on no medications (four male, six female, 22–58 years old). The first 5 mL of blood was discarded to avoid any thrombin that may have been generated around the needle insertion site and then drawn into tubes containing 3.2% v/v sodium citrate. Blood from eight patients at a single center who had ECMO support (four male, four female, 34–67 years old) was drawn into ACD-A tubes (BD Vacutainer). Patients with ECMO support received anti-coagulation and/or anti-platelet medications based on clinicians' discretion or institutional guidelines where patients typically commenced on warfarin, target international normalized ratio (INR) 2–3, with bridging heparin infusion and aspirin therapy, as well as dipyridamole for those who are considered high risk for thrombosis. Plasma was prepared by twice centrifugation at $800 \times g$ for 20 min at room temperature. On some occasions, healthy donor plasma was sheared at rates of $2000 \, s^{-1}$ or $10000 \, s^{-1}$ for 5 min at room temperature using a Kinexus pro+ rheometer.

**Hepatocyte fibrinogen collection.** The human liver cancer cell line HepG2 (ATCC, HB-8065) was cultured in DMEM, 10% foetal calf serum and glutamax at 5% $CO_2$ and 37 °C. Cells at 80–90% confluence were washed with phosphate-buffered saline, incubated for 18 h in DMEM without serum at 5% $CO_2$ and 37 °C or in a hypoxic chamber at 1% $O_2$, 5% $CO_2$ and 37 °C. The conditioned medium was collected (1 mL per $4 \times 10^6$ cells) and the secreted fibrinogen immediately processed as described below.

**Fibrinogen and fibrin assays.** Plasma samples were untreated or incubated with thrombin and/or the fibrin polymerization inhibitor, GPRP (Sigma). Fibrin was prepared in 1 mL of plasma by addition of 50 units of bovine thrombin (Sigma) and 10 mM $CaCl_2$ and $MgCl_2$ for 40 min at 22 °C. The fibrin polymers were collected using a plastic rod. In one experiment, plasma (2 mL) was incubated with 27 Units/mL of thrombin and 10 mM $CaCl_2$ and $MgCl_2$ for 40 min at 22 °C, and

the fibrin polymer collected on a plastic rod and removed. The depleted plasma was incubated with another aliquot of 27 Units/mL of thrombin and the new fibrin polymer collected and removed. An aliquot of plasma (0.2 mL) was sampled before and after addition of both lots of thrombin, 150 μM of D-phenylalanyl-prolyl-arginyl chloromethyl ketone (PPACK, Sigma) added to inhibit thrombin activity and the samples incubated for 16 h at 22 °C. The fibrinogen content of the plasma samples was estimated by collecting the protein on polyclonal anti-fibrinogen antibody-coated beads (see following section), resolving the fibrinogen on SDS-PAGE and staining with colloidal Coomassie. In another experiment, plasma was incubated with 8 mM GPRP (Sigma) for 5 min at 22 °C before addition of 50 units/mL of thrombin for 60 min at 22 °C. The reaction was quenched with 150 μM of PPACK for 10 min at 22 °C. Purified fibrinogens were isolated from healthy donor fresh frozen plasma by β-alanine precipitation[32] or obtained commercially (Sigma F4883).

**Quantification of the redox states of fibrinogen and fibrin disulfide bonds.** Dynabeads (2 mg, Life Technologies) were coated with 16 μg of polyclonal anti-fibrinogen antibodies (Dako, Cat A0080, Lot 00015063) in 1 mL of phosphate-buffered saline on a rotating wheel for 1 h at 22 °C and the excess antibody removed by washing three times with 1 mL of phosphate-buffered saline. Plasma (0.7 mL) or HepG2-conditioned medium (4.5 mL) was incubated with the 2 mg of coated beads on a rotating wheel for 1 h at 22 °C. The beads were collected using a magnet, excess plasma aspirated to reduce the volume, and incubated in 0.3 mL of 5 mM $^{12}$C-IPA in phosphate-buffered saline containing 10% DMSO for 1 h at 22 °C in the dark to alkylate unpaired Cys thiols in the proteins. The supernatants were aspirated, and the beads incubated with NuPAGE LDS sample buffer (Life Technologies) containing a further 5 mM $^{12}$C-IPA for 30 min at 60 °C. The supernatants were resolved on SDS-PAGE. Fibrin polymers generated in plasma were collected on a plastic rod and immediately submerged in 1 mL of 5 mM $^{12}$C-IPA in phosphate-buffered saline containing 10% DMSO for 1 h at 22 °C in the dark. The fibrin was washed 3 times with phosphate-buffered saline before dissolving the polymer in 1 mL of 20 mM acetic acid for 24 h at 22 °C. Twenty microliters of the solution was incubated with NuPAGE LDS sample buffer (Life Technologies) containing 5 mM $^{12}$C-IPA for 30 min at 60 °C and the proteins resolved on SDS-PAGE. Purified fibrinogen (12 μg) was incubated with 5 mM $^{12}$C-IPA in phosphate-buffered saline containing 10% DMSO for 1 h at 22 °C in the dark and the protein resolved on SDS-PAGE.

The SDS-PAGE gels were stained with colloidal coomassie (Sigma) and the fibrin(ogen) bands excised, destained, dried, incubated with 40 mM dithiothreitol and washed[14]. The gel slices were incubated with 5 mM $^{13}$C-IPA (Cambridge Isotopes) for 1 h at 22 °C in the dark to alkylate the disulfide bond Cys, washed and dried before digestion of proteins with 12.5 ng/μL of trypsin (Promega) in 25 mM $NH_4CO_2$ overnight at 25 °C. Peptides were eluted from the slices with 5% formic acid, 50% acetonitrile. Nano-Liquid chromatography (nano-LC) was performed using an Ultimate 3000 HPLC and autosampler system (Dionex). Samples were injected into a fritless nano-LC column (75 μm x ~12 cm) containing C18 media (1.9 μm, 120 Å ReproSil-Pur 120 C18-AQ, Dr Maisch GmbH) and heated to 45 °C for all runs. Peptides were eluted using a linear gradient over 52 min, at a flow rate of 0.2 μL/min. Mobile phase A consisted of 0.1% formic acid in $H_2O$, while mobile phase B consisted of acetonitrile:$H_2O$ (8:2) with 0.1% formic acid. The gradient was: 0 min 2% B; 4 min 2% B, 36 min 45% B, 37.5 min 80% B, 39 min 2% B and 52 min 2% B. High voltage (2000 V) was applied to a low volume tee (Upchurch Scientific) and the column tip positioned ~0.5 cm from the heated capillary ($T = 275$ °C) of a LTQ Orbitrap Velos (Thermo Electron) mass spectrometer. Positive ions were generated by electrospray ionization and the mass spectrometer operated in data dependent acquisition mode. Full scan MS spectra were acquired ($m/z$ 350–1750) by the Orbitrap at a resolution of 30,000. The 15 most abundant ions (>5000 counts) with charge states ≥ +2 were sequentially isolated and fragmented within the linear ion trap using collisionally induced dissociation with an activation $q = 0.25$ and activation time of 10 ms at a target value of 30,000 ions. $M/z$ ratios selected for MS/MS were dynamically excluded for 35 s. Ion Trap Zoom AGC Target: 3000.00. Ion Trap Full AGC Target: 30000.00. Ion Trap SIM AGC Target: 10000.00. Ion Trap MSn AGC Target: 10000.00. Data were analyzed using Mascot Daemon (Version 2.5.0, Matrix Science) against Swissprot database. Search parameters were precursor tolerance of 6 ppm and product ion tolerances of ±0.6 Da, and iodoacetanilide derivative (Cys), iodoacetanilide $^{13}$C derivative (Cys), oxidation (Met) selected as variable modifications with full tryptic cleavage of up to three missed cleavages. With the list of cysteine-containing peptides identified using Mascot, their monoisotopic mass/charge ratio ($m/z$) of +2, +3, and +4 are calculated using MS-Product (Version v 6.2.2, Protein Prospector Tools). Cys labeled with $^{12}$C-IPA or $^{13}$C-IPA has a mass of 133.05276 or 139.07289, respectively. Thirty Cys-containing peptides were analyzed (Supplementary Tables 1 and 2). The different redox forms of the Cys residues were quantified from the relative ion abundance of peptides labeled with $^{12}$C-IPA and/or $^{13}$C-IPA. To calculate ion abundance of peptides, extracted ion chromatograms were generated using XCalibur Qual Browser (v2.1.0; Thermo Scientific). The area was calculated using the automated peak detection function built into the software. The data was routinely searched for peptides containing free

Cys thiols and these were not detected, which indicates that alkylation of unpaired Cys residues by $^{12}$C-IPA or $^{13}$C-IPA was complete in the protein.

**Fibrinogen disulfide-linked peptide analysis**. Healthy donor plasma (0.7 mL) was incubated with polyclonal anti-fibrinogen antibody-coated Dynabeads on a rotating wheel for 1 h at 22 °C. The beads were collected, excess plasma aspirated and 0.3 mL of 5 mM $^{12}$C-IPA in phosphate-buffered saline containing 10% DMSO was added and incubated for 1 h at 22 °C in the dark. Then the supernatant was aspirated, the beads incubated with NuPAGE LDS sample buffer for 30 min at 60 °C and the supernatants resolved on SDS-PAGE. Gel slices containing the fibrinogen were washed and dried before digestion with 12 ng/μL trypsin (Promega) for 4 h at 37 °C. Reactions were stopped by adding 5% (v/v) formic acid and peptides eluted from the gel slices with 5% formic acid and 50% (v/v) acetonitrile. Peptides in 0.1% formic acid (final volume 12 μL) were resolved on a 35 cm × 75 μm C18 reverse phase analytical column using a 2–35% acetonitrile gradient over 22 min at a flow rate of 300 nL/min (Thermo Fisher Scientific Ultimate 3000 HPLC) and analyzed on a LTQ Orbitrap Velos mass spectrometer as described above. Disulfide-linked peptides were searched against human fibrinogen sequence using Byonic analysis software (Version 3.9-7, Protein Metrics) with a false discovery rate of 1%. All disulfide-linked peptides were manually inspected for accuracy. Precursor mass tolerance and fragment tolerance were set at 10 ppm and 0.6 Da, respectively. Variable modifications were defined as oxidized Met, oxidized Cys (mono, di and tri) and glutathionylated Cys with full trypsin cleavage of up to three missed cleavages.

**Fibrinogen functionality assay**. Purified fibrinogen (0.2 mL of 10 mg/mL in 50 mM Hepes, 0.14 M NaCl, 10 mM CaCl$_2$, pH 7.4) was incubated with 5 mM $^{12}$C-IPA for 1 h at 22 °C in the dark to alkylate unpaired Cys thiols in the protein. The unreacted IPA was removed using a 7 kDa MWCO Zeba spin desalting column (Thermo Fisher). Fibrinogen or fibrinogen-IPA (1.1 mg/mL in 50 mM Hepes, 0.14 M NaCl, 10 mM CaCl$_2$, pH 7.4) was incubated with 1 unit/mL of bovine thrombin (Sigma) for 5 h at 22 °C in a total volume of 0.2 mL in 1.5 mL microcentrifuge tubes. The fibrin polymers were pelleted at 13,000 × $g$ for 15 min and the concentration of fibrinogen remaining in the supernatant measured. Functionality is expressed at the % of fibrinogen consumed in fibrin polymer formation.

**Fibrin polymer formation measured by light scattering**. Purified fibrinogen or fibrinogen-IPA (1 mg/mL in 50 mM Hepes, 0.14 M NaCl, 10 mM CaCl$_2$, pH 7.4) was incubated with 1 unit/mL of bovine thrombin (Sigma). The increase in absorbance at 350 nm was recorded every 30 s for 15 min.

**Fibrin polymer structure measured by scanning electron microscopy (SEM)**. Fibrinogen or fibrinogen-IPA (1 mg/mL in 50 mM Hepes, 0.14 M NaCl, 10 mM CaCl$_2$, pH 7.4) was incubated with 1 unit/mL of bovine thrombin (Sigma) for 2 h at 22 °C in a total volume of 0.2 mL. The fibrin polymers were rinsed twice with 0.1 M phosphate-buffer saline (PBS), fixed with 2.5% glutaraldehyde in PBS overnight at 4 °C, post-fixed with 1% OsO$_4$ in PBS, dehydrated with graded series of ethanol, and next critical point dried with a Leica EM CPD300. Samples were sputter coated with 15 nm gold (K550X, Emitech) and imaged with a Zeiss SigmaHD Field Emission Scanning Electron Microscope. The SEM micrographs were recorded at 5 kV accelerating voltage and a working distance of 5 mm. Fiber diameters were measured using ImageJ.

**Fibrin polymer permeation assay**. Fibrinogen or fibrinogen-IPA (1 mg/mL in 50 mM Hepes, 0.14 M NaCl, 10 mM CaCl$_2$, pH 7.4) was incubated with 1 unit/mL of bovine thrombin (Sigma) for 2 h at 22 °C in a total volume of 0.2 mL in polypropylene chromatography columns with an internal diameter 4.6 mm. The columns were overlayed with 0.8 mL of the Hepes buffer and flux ($J$) was calculated from the weight of the drops that percolated in 20 min. The permeation (Darcy) constant[33] was calculated from the relationship, $Ks = J\eta A/LP$, where $J$ is the flux (cm$^3$/s), $\eta$ is the viscosity of the buffer (0.01 poise at room temperature), $A$ is the cross-section of the clot (0.17 cm$^2$), $L$ is the length of the clot (1.1 cm), and $P$ is the hydrostatic pressure (1,018,218 dyne/cm$^2$).

**Fibrin polymer compaction assay**. Fibrinogen or fibrinogen-IPA (1 mg/mL in 50 mM Hepes, 0.14 M NaCl, 10 mM CaCl$_2$, pH 7.4) was incubated with 1 unit/mL of bovine thrombin (Sigma) for 16 h at 22 °C in a total volume of 0.5 mL in 1.5 mL microcentrifuge tubes previously coated with 10 mg/mL PEG-20000 in water and air dried. The fibrin polymers were centrifuged at 13,000 × $g$ for 5 to 2400 s and the volume of the fluid extruded from the polymer measured.

**Fibrinolysis assay**. Fibrinogen or fibrinogen-IPA (1.8 mg/mL in 50 mM Hepes, 0.14 M NaCl, 10 mM CaCl$_2$, pH 7.4) was incubated with 1 unit/mL of bovine thrombin (Sigma) for 2 h at 22 °C in a total volume of 0.5 mL in 24-well plates (15.6 mm diameter) for 2 h at 22 °C. An aliquot of plasmin (5 μL of 0.59 μM) was added to the center of the wells and the plates incubated for 18 h at 37 °C in a humid environment. Lysis areas were determined from the average of two diameters measured at right angles.

**Quantification of the redox states of α2-macroglobulin disulfide bonds**. The 10 healthy donor plasmas described above for analysis of fibrinogen were employed for analysis of α2-macroglobulin using the same procedures. Plasma (0.7 mL) was incubated with polyclonal anti-α2-macroglobulin antibody (Dako, Cat Q0102, Lot 20068567)-coated Dynabeads (Life Technologies) on a rotating wheel for 1 h at 22 °C. The beads were collected, excess plasma aspirated, and incubated in 0.3 mL of 5 mM $^{12}$C-IPA in phosphate-buffered saline containing 10% DMSO for 1 h at 22 °C in the dark to alkylate unpaired Cys thiols in the proteins. The supernatants were aspirated, and the beads incubated with NuPAGE LDS sample buffer (Life Technologies) containing a further 5 mM $^{12}$C-IPA for 30 min at 60 °C. The supernatants were resolved on SDS-PAGE. The SDS-PAGE gels were stained with colloidal coomassie (Sigma) and the α2-macroglobulin band excised, destained, dried, incubated with 40 mM dithiothreitol and washed[14]. The gel slices were incubated with 5 mM $^{13}$C-IPA (Cambridge Isotopes) for 1 h at 22 °C in the dark to alkylate the disulfide bond Cys, washed and dried before digestion of proteins with 12.5 ng/μl of trypsin (Promega) in 25 mM NH$_4$CO$_2$ overnight at 25 °C. Peptides were eluted from the slices with 5% formic acid, 50% acetonitrile. Liquid chromatography, mass spectrometry and data analysis were performed as described for fibrinogen[34]. 17 Cys-containing peptides were analyzed (Supplementary Tables 2 and 4). The different redox forms of the Cys residues were quantified from the relative ion abundance of peptides labeled with $^{12}$C-IPA and/or $^{13}$C-IPA.

**Reporting summary**. Further information on research design is available in the Nature Research Reporting Summary linked to this article.

## Data availability

The mass spectrometry proteomics data have been deposited to the ProteomeXchange Consortium via the PRIDE [1] partner repository with the dataset identifier "PXD018564". Protein Data Bank (PDB) identifiers "3ghg [https://doi.org/10.2210/pdb3GHG/pdb]" and "4acq [https://doi.org/10.2210/pdb4ACQ/pdb]" were used. Any other relevant data are available from the authors. Source data are provided with this paper.

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

## Acknowledgements

We thank Joyce Chiu for assistance with the disulfide-linked peptide analysis, Elizabeth Gardiner, Amanda Davis, Deirdre Murphy and Robert Andrews for the heart failure patient plasmas, Marc Ellis for assistance with shearing of plasma, and Aster Pijning and Manuela Florido for helpful discussions. Mass spectrometric results were obtained at the Bioanalytical Mass Spectrometry Facility within the Mark Wainwright Analytical Center of the University of New South Wales.

## Author contributions

P.J.H. conceived the study, designed experiments, and wrote the manuscript. D.B. designed and performed the experiments.

## Competing interests

The authors declare no competing interests.
