## [Peer Review File · Nature Communications]

Reviewers' comments:

Reviewer #1 (Remarks to the Author):

This manuscript provides a comprehensive exploration of the redox state of disulfide bonds on the circulating protein fibrinogen. The data provided supports the conclusion that each individual disulfide exists in a mixture of reduced and oxidation states, and there is little variability in the percentage of each state across different patients. The distribution of disulfide bonds changes upon fibrin formation, shear forces, and cardiovascular disease state. The general observation that multiple redox states can exist across disulfides within a single protein is shown to be true for a second circulating protein, VWF.

Together, these data provide intriguing evidence to support the hypothesis that native proteins can exist in multiple disulfide-bonded states. The data provided are fully supportive of their conclusions, and I think this manuscript will be of interest to a broad audience. I recommend publication upon addressing the minor comments below:

1. In the text and figure legend, 14 disulfide bonds in fibrinogen are discussed, but only 13 are shown in Figure 1C.
2. The authors state that "6 of the nine D region disulfides in fibrin were >30% more oxidized than the same bonds in plasma fibrinogen". However, the data in Figure 3B does not show greater than 30% change for any of the disulfides - all the changes are to a much lesser degree.
3. The authors claim that "a subset of fibrinogen disulfide-bonded states containing more oxidized nodules is preferentially used for fibrin fibre formation". To make this claim, the authors need to show data showing that the more oxidized fibrinogen is concurrently depleted from the soluble pool.
4. For the patients with heart failure, only 4 patient samples were analyzed. Given that this is a small number of samples, the authors should be cautious about any claims made in this section, and should clearly state that this was a small number of patient samples that makes true statistical analysis difficult.

5. In the discussion, the authors state that "our preliminary findings indicate that the protease inhibitor, alpha-2 macroglobulin, and the zymogen, prothrombin, also exist in multiple covalent states...". However, no data is provided on these two proteins, so this claim is not substantiated in this current manuscript and should be removed.

6. Similarly, the authors state that..."it is not appreciably influenced by donor age or gender". A table separating the donors by age/gender, together with the corresponding redox profiles need to be provided to support this statement.

7. In addition to the raw mass spectrometry data that the authors have provided, it would be helpful to include a supplementary table that lists the isotopic ratios obtained for each cysteine-containing peptide in each of the individual patient samples.

Reviewer #2 (Remarks to the Author):

Butera et al. used a mass spectrometry-based approach to quantify the redox state of disulfide bonds in the blood proteins Fibrinogen and von Willebrand Factor (vWF). The individual disulfides showed remarkably variable oxidation levels, with disulfide reduction levels between 3 and 50%. The authors conclude that those proteins might exist in hundreds of different disulfide states in human blood, and their results indicate that these states change for specific disulfides, e.g. upon sheer stress as well as under pathophysiological conditions. They conclude that these different disulfide states modulate protein function and that they furthermore have important implications e.g. for drug design and drug treatment.

While generally interesting the manuscript raises a couple of questions. The methods part is very superficial, and the data presented is not too comprehensive. The title is misleading and has to be changed, as the authors rather speculate but do not demonstrate that the different disulfide oxidation states indeed lead to different functions, as the title indicates. In general it is intriguing how reproducible the measurements of disulfide reaction states from blood proteins are --- especially for the diseased samples, as in my experience diseased samples tend to be even more variable than healthy controls.

Major comments:

1) How was the blood collected in detail? Small needle diameters may induce platelet activation, which in turn triggers release of vWF from alpha granules. This could potentially affect the results and might explain the more variable levels of oxidation compared to the Fibrinogen.

2) A protein-IP based workflow might already induce some bias (even if polyclonal) and enrich only certain subclasses of the target protein, especially if it exists in such diverse PTM patterns as the authors mention in their manuscript. It would be good to have a control to ensure that their targets were efficiently pulled-out, for instance using targeted MS assays against a few representative peptides of those proteins, that could be applied after IP, or at least using a good Western blot antibody. Details for the IP process are basically completely missing, and the protocol cannot be reproduced this way (how much antibody, what buffer, what total volume, etc.).

3) Is it essential to control that the blocking of free Cys was indeed complete. The authors mention that they have a second step of blocking to ensure complete blocking, but do not show any controls that demonstrate this. Also, the entire workflow is rather slow and under alkaline conditions disulfide scrambling can occur. The authors should comment on that. This could include glutathionylation – how can it be excluded that some of the potential disulfides are actually glutathionylated Cys, as these can also be reduced with DTT? Since peptides are not detected as intact disulfides, how can the authors make sure that what they detect was really originating from the anticipated disulfide bonds indicated in some of the plots --- as the authors themselves point out there might be many more disulfide bonds and as they also mention, data from crystallization might be misleading or incomplete. On a similar note, the protocol for fibrin production and analysis also sounds prone to artifacts, how did the authors control that disulfide states are not influenced by the sample preparation?

4) The statement that the redox states of disulfide bonds would have been ‘modeled’ sounds a bit exaggerated, as modeling usually includes thorough computational calculations, whereas the data presented is based on simple counting. For instance including energy states and/or distances. For instance, it is unclear how many of the disulfide states in figure 2D are even sterically possible or energetically favorable. Later, the authors also mention that theoretically 16,384 disulfide-states would be possible for Fibrinogen and that this might be an overestimation --- again, a real modelling (e.g. including structural constraints, etc.) would probably already rule out the vast majority of these combinations. The statement that “disulfide formation is conditional” does not sound surprising in this context.

5) The statement that a subset of fibrinogen disulfide-bonded states is used for fibrin formation is bold, the experimental data does not allow such a statement, this would require additional confirmation.

6) The authors mention that they had preliminary data for further proteins to support their claims. Why was this data not included in the present manuscript? It generally does not contain a lot of data, and no data is provided to actually support the claim that there might be hundreds of disulfide states with functional differences. Though entirely possible, this statement would need some more demonstration.

Minor comments:

1) In the method part it is not clear how many and what samples have been sheared and how. All experiments are described rather superficially, and cannot be reproduced this way. The MS part is

just referring to another manuscript and does not contain any information. Even when citing their own previous work, the authors should provide a thorough protocol, as a protocol is rarely 100% the same and the readers (and reviewers) should be able to understand what has been done and exactly how. This includes also the database search and data analysis, all very unclear.

2) It is unclear why the authors used a rather unusual alkylation agent, I guess it is to introduce a minimum mass shift for the heavy labeled version? How well has it been characterized that IPA leads to full blocking of free Cys under the used conditions?

3) The authors mention that peptides were quantified by HPLC and identified by MS (the wording used is actually a bit odd) --- how were peptides quantified by HPLC, that is unclear? And how was the loading amount controlled? Indeed the samples I checked as RAW files look surprisingly reproducible in terms of ion intensities.

4) "The reduced species of each individual disulfide was summed and this number expressed as percentage of the total number of disulfide-bonded states to mirror the experimental form of the data" --- What does that mean, it is not clear to me.

5) From the text it is unclear how many replicates were used for the shear stress analysis.

Reviewer #3 (Remarks to the Author):

The authors have investigated the redox state of disulfide bonds in the plasma proteins fibrinogen and von Willebrand factor (VWF). They conclude that these proteins exist in different disulfide-bonded states in the circulation which can be influenced by cardiovascular disease.

On page 1 it is mentioned that "Isolating a protein that contains closely spaced cysteine thiols and storing it in ambient oxygen can result in oxidation of the thiols to a disulfide bond. To exclude this possibility, we froze the redox state of cysteine thiols before the proteins were removed from their native environment, which in this case was blood plasma". However, freezing of the redox state was performed about 2 hours after the blood draw with significant transferring of blood and plasma from one container to another, centrifugation steps, rotating incubation and shearing during which the proteins were likely to get into contact with ambient oxygen. Why was the incubation with 12C-IPA not performed earlier for the samples without shearing, e.g. right during/after blood draw? Controls should be performed to check whether the redox state is altered during all these processing steps.

Please discuss the difference in disulfide reduction in fibrinogen from sheared normal plasma compared to the fibrinogen from patients with cardiac failure. What is the suggested mechanism of the observed reduction of disulfide bonds in the circulation?

The statement: "To test whether the multiple disulfide-bonded states of fibrinogen is a property of other circulating proteins, we examined von Willebrand factor (VWF)" is misleading since the authors and other groups have already previously shown varying disulfide bond states and disulfide switches under shear conditions for VWF, just in other domains of VWF. In one publication, the VWF redox state was also investigated in patients with heart failure. These publications should be cited and discussed in the context of this manuscript.

Reviewer #1

1. *In the text and figure legend, 14 disulfide bonds in fibrinogen are discussed, but only 13 are shown in Figure 1C.*

This has been corrected in the revised manuscript. The redox state of 14 of the 17 fibrinogen disulfides was quantified: 5 in the E region and 9 bonds in the D region. Of the 14 bonds that were measured, 13 have been structurally defined. The D region α C491 is predicted to pair with α C461, although this bond has not been structurally characterised and has been omitted from this analysis. See page 2, paragraph 4 of the revised manuscript.

2. *The authors state that "6 of the nine D region disulfides in fibrin were >30% more oxidized than the same bonds in plasma fibrinogen". However, the data in Figure 3B does not show greater than 30% change for any of the disulfides - all the changes are to a much lesser degree.*

We have omitted this description of relative degree of change in redox state in the revised manuscript. We now simply state that all 13 disulfides are more oxidized in fibrin polymer: $p < 0.05$ for the α C68- β C106 bond and $p < 0.01$ for the other 12 bonds (see **Fig. 3**, page 14).

3. *The authors claim that "a subset of fibrinogen disulfide-bonded states containing more oxidized nodules is preferentially used for fibrin fibre formation". To make this claim, the authors need to show data showing that the more oxidized fibrinogen is concurrently depleted from the soluble pool.*

We performed this experiment and the results led us to revise our conclusions. We showed that all 13 disulfides are significantly more oxidized in fibrin polymer (see **Fig. 3B**). We considered two scenarios that could account for this change: 1) a subset of fibrinogen covalent states containing more oxidized disulfide bonds is preferentially used for fibrin formation, or 2) disulfide bonds form during fibrin formation. The first scenario implies there would be enrichment of reduced fibrinogen states as fibrin polymer forms in plasma, as indicated by the reviewer. This was tested by adding a limiting concentration of thrombin to healthy donor plasma, the fibrin polymer removed, another aliquot of thrombin added to the depleted plasma and the new fibrin polymer removed. Fibrinogen concentration in the plasma was depleted 41% in the first reaction and >95% in the second reaction. There was no change in the disulfide-bonded states of fibrinogen as fibrin formed (see new **Fig. 4A**), which supports the second scenario that disulfide bonds oxidise during fibrin formation. We next determined whether disulfides form before or after fibrin polymerises. Thrombin was added to healthy donor plasmas and the resulting fibrin was prevented from polymerising by the tetrapeptide inhibitor, GPRP. GPRP binds to the γ -nodule 'hole' and blocks fibrin binding. There is no oxidation of disulfide bonds in the fibrin if polymerisation is prevented (see new **Fig. 4B**), indicating that disulfides form as a result of fibrin polymerisation. See page 3, paragraphs 4 and 5 of the revised manuscript.

4. *For the patients with heart failure, only 4 patient samples were analyzed. Given that this is a small number of samples, the authors should be cautious about any claims made in this section, and should clearly state that this was a small number of patient samples that makes true statistical analysis difficult.*

This data has been omitted from the revised manuscript due to the small numbers. We have instead compared the effects of *in vitro* pathological fluid shear on fibrinogen covalent states with a clinical situation associated with high blood shear. This situation is treatment of refractory heart failure patients with extracorporeal membrane oxygenation (ECMO) support. The ECMO device subjects blood to high shear and both thrombosis and bleeding can occur with this intervention. Eight of the 13 fibrinogen disulfide bonds were significantly ($p < 0.001$) more reduced in patients receiving ECMO ($n = 8$) than in healthy human donors ($n = 10$) (see new **Fig. 6B**). These differences are notable considering the variability in disease severity and therapeutic intervention in these patients and suggest that reduction of the fibrinogen disulfides is a result of the blood shear associated with the device. See page 5, paragraph 2 of the revised manuscript.

5. In the discussion, the authors state that "our preliminary findings indicate that the protease inhibitor, alpha-2 macroglobulin, and the zymogen, prothrombin, also exist in multiple covalent states...". However, no data is provided on these two proteins, so this claim is not substantiated in this current manuscript and should be removed.

We have included the α 2-macroglobulin data in the revised manuscript and deleted this statement from the discussion. α 2-Macroglobulin is a broad-spectrum endopeptidase inhibitor produced by the liver and macrophages and circulates in blood. It regulates proteolysis in several biological processes, including nutrition, tissue remodelling and infection. As we observed for fibrinogen, α 2-macroglobulin also exists as multiple disulfide-bonded states in the circulation. The enzyme contains 12 structurally-defined disulfide bonds and we were able to quantify the redox state of all 12. The 12 bonds range from 10 to 70% reduced in eight healthy human donors (3 male, 5 female, 18-48 years old) (see new **Tables S4** and **S5** and new **Fig. S4**). See page 5, paragraph 3 of the revised manuscript.

6. Similarly, the authors state that..."it is not appreciably influenced by donor age or gender". A table separating the donors by age/gender, together with the corresponding redox profiles need to be provided to support this statement.

This gender data has been included in revised **Fig. 1D**. See page 12 of the revised manuscript.

7. In addition to the raw mass spectrometry data that the authors have provided, it would be helpful to include a supplementary table that lists the isotopic ratios obtained for each cysteine-containing peptide in each of the individual patient samples.

This data has been included in the revised manuscript. Supplementary **Table S2** lists the HPLC retention times of fibrinogen peptides, ^{12}C -IPA- and ^{13}C -IPA-labelled peptide AUC values and isotopic ratios for the ten healthy donor proteins. See pages 19-23 of the revised manuscript. This data has for analysis of α 2-macroglobulin has also been included (see **Table S5**). See pages 25-28 of the revised manuscript.

Reviewer #2

Major comments:

1) How was the blood collected in detail? Small needle diameters may induce platelet activation, which in turn triggers release of vWF from alpha granules. This could potentially affect the results and might explain the more variable levels of oxidation compared to the Fibrinogen.

The details of blood collection have been included in the revised manuscript. Measurement of plasma versus platelet VWF is a valid concern. In addition, our analysis of VWF covers only a small fraction of the disulfide bonds in the protein and we are relying on disulfide pairings from crystal structures of subdomains of VWF, not the full-length molecule. For these reasons, we have omitted the VWF results from the revised manuscript and replaced them with analysis of plasma α 2-macroglobulin. α 2-Macroglobulin is a broad-spectrum endopeptidase inhibitor produced by the liver and macrophages and circulates in blood. It regulates proteolysis in several biological processes, including nutrition, tissue remodelling and infection. As we observed for fibrinogen, α 2-macroglobulin also exists as multiple disulfide-bonded states in the circulation. The enzyme contains 12 structurally-defined disulfide bonds and we were able to quantify the redox state of all 12. The 12 bonds range from 10 to 70% reduced in eight healthy human donors (3 male, 5 female, 18-48 years old) (see new **Tables S4** and **S5** and new **Fig. S4**). See page 5, paragraph 3 of the revised manuscript.

2) A protein-IP based workflow might already induce some bias (even if polyclonal) and enrich only certain subclasses of the target protein, especially if it exists in such diverse PTM patterns as the authors mention in their manuscript. It would be good to have a control to ensure that their targets were efficiently pulled-out, for instance using targeted MS assays against a few representative

peptides of those proteins, that could be applied after IP, or at least using a good Western blot antibody. Details for the IP process are basically completely missing, and the protocol cannot be reproduced this way (how much antibody, what buffer, what total volume, etc.).

We apologise for the lack experimental details and have corrected this in the revised manuscript. See pages 6-8 of the revised manuscript.

The question of whether the immunoprecipitated fibrinogen is representative of the total fibrinogen pool has been addressed in two ways. We have compared the redox state of fibrinogen disulfide bonds in protein immunoprecipitated from plasma versus protein purified from plasma using β -alanine precipitation. While there is a degree of oxidation of disulfide bonds in fibrinogen when it is removed from plasma, the redox state of the bonds had the same pattern as for the immunoprecipitated protein (see new **Fig. S2**). In another experiment, a limiting concentration of thrombin was added to healthy donor plasma, the fibrin polymer removed, another aliquot of thrombin added to the depleted plasma and the new fibrin polymer removed. Fibrinogen concentration in the plasma was depleted 41% in the first reaction and >95% in the second reaction. There was no change in the disulfide-bonded states of the successively depleted fibrinogen pools (see new **Fig. 4A**). Notably, all the fibrinogen in the second plasma pool was captured by immunoprecipitation, so this data reflects the total fibrinogen pool in that sample. In summary, there is no indication that our results do not reflect the total plasma fibrinogen pool.

3) Is it essential to control that the blocking of free Cys was indeed complete. The authors mention that they have a second step of blocking to ensure complete blocking, but do not show any controls that demonstrate this. Also, the entire workflow is rather slow and under alkaline conditions disulfide scrambling can occur. The authors should comment on that. This could include glutathionylation – how can it be excluded that some of the potential disulfides are actually glutathionylated Cys, as these can also be reduced with DTT? Since peptides are not detected as intact disulfides, how can the authors make sure that what they detect was really originating from the anticipated disulfide bonds indicated in some of the plots --- as the authors themselves point out there might be many more disulfide bonds and as they also mention, data from crystallization might be misleading or incomplete. On a similar note, the protocol for fibrin production and analysis also sounds prone to artifacts, how did the authors control that disulfide states are not influenced by the sample preparation?

Free cysteines were blocked with ^{12}C -IPA in the native protein and then in the SDS-denatured protein in Laemmli buffer. The concentration (5 mM) and time of incubation with ^{12}C -IPA (60 min) was maximised to achieve complete labelling. Our mass spectrometry data was routinely searched for peptides containing free Cys thiols and these were not detected. See page 8, paragraph 1 of the revised manuscript. Disulfide scrambling is prevented by blocking free Cys thiols. We certainly agree that assigning the correct disulfide bond pairing in fibrinogen is a key factor in our studies. This is the reason why we chose to characterise only those fibrinogen and α 2-macroglobulin disulfide bonds that have been structurally defined, which is the most definitive evidence currently. Note that the disulfide pairing for these proteins is consistent in different 3D structures (e.g., compare **Figs. 1** and **S5** for the D domain fibrinogen disulfides), which is reassuring. In the revised manuscript, we have sought to provide additional evidence for the disulfide pairing in fibrinogen from mass spectrometry analysis of disulfide-linked peptides. Fibrinogen was immunoprecipitated from a healthy 25-year-old male donor plasma, digested with trypsin and peptides resolved by HPLC and analysed by mass spectrometry. Thirteen disulfide-linked peptides encompassing 7 of the 14 disulfides measured in the redox state analysis were identified (see new **Table S3**), which is in accordance with the disulfide pairing observed in crystal structures. See page 2, paragraph 5 of the revised manuscript. We also searched for glutathionylated Cys-containing peptides in this data set and these were not found. Regarding analysis of fibrin, the polymer was generated in plasma and immediately quenched with 5 mM ^{12}C -IPA to freeze free thiols. The fibrin polymer was then dissociated in acetic acid, which would protonate any cysteine thiols that may not have been blocked with ^{12}C -IPA and prevent possible

thiol/disulfide exchange events. We made every effort to eliminate processing artefacts in our experiments and there are no indications in our data of experimental artefacts. Nevertheless, if there was a small degree of disulfide oxidation during processing this would not change the conclusions of our study.

4) The statement that the redox states of disulfide bonds would have been 'modeled' sounds a bit exaggerated, as modeling usually includes thorough computational calculations, whereas the data presented is based on simple counting. For instance including energy states and/or distances. For instance, it is unclear how many of the disulfide states in figure 2D are even sterically possible or energetically favorable. Later, the authors also mention that theoretically 16,384 disulfide-states would be possible for Fibrinogen and that this might be an overestimation --- again, a real modelling (e.g. including structural constraints, etc.) would probably already rule out the vast majority of these combinations. The statement that "disulfide formation is conditional" does not sound surprising in this context.

We have replaced the term 'modelling' with 'simulation' in the revised manuscript.

5) The statement that a subset of fibrinogen disulfide-bonded states is used for fibrin formation is bold, the experimental data does not allow such a statement, this would require additional confirmation.

We have performed additional experiments and the results have led us to revise our conclusions. We showed that all 13 disulfides are significantly more oxidized in fibrin polymer (see **Fig. 3B**). We considered two scenarios that could account for this change: 1) a subset of fibrinogen covalent states containing more oxidized disulfide bonds is preferentially used for fibrin formation, or 2) disulfide bonds form during fibrin formation. The first scenario implies there would be enrichment of reduced fibrinogen states as fibrin polymer forms in plasma, as indicated by the reviewer. This was tested by adding a limiting concentration of thrombin to healthy donor plasma, the fibrin polymer removed, another aliquot of thrombin added to the depleted plasma and the new fibrin polymer removed. Fibrinogen concentration in the plasma was depleted 41% in the first reaction and >95% in the second reaction. There was no change in the disulfide-bonded states of fibrinogen as fibrin formed (see new **Fig. 4A**), which supports the second scenario that disulfide bonds oxidise during fibrin formation. We next determined whether disulfides form before or after fibrin polymerises. Thrombin was added to healthy donor plasmas and the resulting fibrin was prevented from polymerising by the tetrapeptide inhibitor, GPRP. GPRP binds to the γ -nodule 'hole' and blocks fibrin binding. There is no oxidation of disulfide bonds in the fibrin if polymerisation is prevented (see new **Fig. 4B**), indicating that disulfides form as a result of fibrin polymerisation. See page 3, paragraphs 4 and 5 of the revised manuscript.

6) The authors mention that they had preliminary data for further proteins to support their claims. Why was this data not included in the present manuscript? It generally does not contain a lot of data, and no data is provided to actually support the claim that there might be hundreds of disulfide states with functional differences. Though entirely possible, this statement would need some more demonstration.

The title has been changed to specifically refer to fibrinogen and the revised manuscript contains several additional data sets in support of the functional claims (new **Figs. 4, 5, 6, S2, S3, S4, S6**). We have also included our analysis of the α 2-macroglobulin disulfide bonds in the revised manuscript (see Point 1).

Minor comments:

1) In the method part it is not clear how many and what samples have been sheared and how. All experiments are described rather superficially, and cannot be reproduced this way. The MS part is just referring to another manuscript and does not contain any information. Even when citing their

own previous work, the authors should provide a thorough protocol, as a protocol is rarely 100% the same and the readers (and reviewers) should be able to understand what has been done and exactly how. This includes also the database search and data analysis, all very unclear.

In the revised manuscript, we have referenced a recent detailed protocol published by the team (Chiu J. Quantification of the redox state of protein disulfide bonds. *Methods Mol Biol* 1967, 45-63, 2019; reference 31), which is exactly the protocol that was followed in this study.

2) *It is unclear why the authors used a rather unusual alkylation agent, I guess it is to introduce a minimum mass shift for the heavy labeled version? How well has it been characterized that IPA leads to full blocking of free Cys under the used conditions?*

We have tested other pairs of alkylating agents in the past, such as N-ethylmaleimide and iodoacetamide, but have found the ^{12}C -IPA/ ^{13}C -IPA pair to be the best in our hands for quantifying the redox state of protein disulfide bonds. They are thiol-specific alkylators with the same effective size and same efficiency of alkylation of unpaired cysteines. That is, they have the same access to and same rate of reaction with cysteines, which is important for producing quantitative data. Re full blocking, see Point 3.

3) *The authors mention that peptides were quantified by HPLC and identified by MS (the wording used is actually a bit odd) --- how were peptides quantified by HPLC, that is unclear? And how was the loading amount controlled? Indeed the samples I checked as RAW files look surprisingly reproducible in terms of ion intensities.*

We apologise for this confusion. The fraction of reduced disulfide was calculated from the ratio of the peak AUC of the ^{12}C -IPA-labelled Cys-containing peptide divided by the sum of the peak AUC's of the ^{12}C -IPA-labelled and ^{13}C -IPA-labelled peptides. See **Fig. S1** and **Table S2**. The measure of disulfide redox state for a given sample is calculated from the internal ^{12}C - versus ^{13}C -labelled peptide ratios, so is independent of differences in loading amount from sample to sample.

4) *“The reduced species of each individual disulfide was summed and this number expressed as percentage of the total number of disulfide-bonded states to mirror the experimental form of the data” --- What does that mean, it is not clear to me.*

The experimental data is expressed as % of reduced disulfide bond in the population of protein molecules. The simulated numbers have been expressed in the same way to reflect the experimental data. This description has been improved in the revised manuscript. See page 3, paragraph 2 of the revised manuscript.

5) *From the text it is unclear how many replicates were used for the shear stress analysis.*

Four different healthy donor were plasmas sheared at rates of 2000s^{-1} or 10000s^{-1} for 5 min (see revised **Fig. 6**).

Reviewer #3

On page 1 it is mentioned that “isolating a protein that contains closely paced cysteine thiols and storing it in ambient oxygen can result in oxidation of the thiols to a disulfide bond. To exclude this possibility, we froze the redox state of cysteine thiols before the proteins were removed from their native environment, which in this case was blood plasma”. However, freezing of the redox state was performed about 2 hours after the blood draw with significant transferring of blood and plasma from one container to another, centrifugation steps, rotating incubation and shearing during which the proteins were likely to get into contact with ambient oxygen. Why was the incubation with ^{12}C -IPA not performed earlier for the samples without shearing, e.g. right during/after blood draw? Controls should be performed to check whether the redox state is altered during all these processing steps.

This is a fair point. We have changed our description of the protein preparation in the revised manuscript as follows: ‘The redox state of cysteine thiols in fibrinogen were frozen before the

protein was removed from its *ex vivo* plasma environment.' Ideally, we would add ^{12}C -IPA to freshly drawn blood but it is not practical to achieve 5 mM concentration in a large volume of blood. In the revised manuscript, we have measured the effect of isolating fibrinogen from plasma on the redox state of the disulfide bonds. Two preparations of purified fibrinogen were tested, one that was stored frozen at -20°C and another that was stored lyophilised. The disulfide bonds of the frozen preparation were more oxidized than healthy donor fibrinogens prepared directly from plasma (see new **Fig. S2**). The lyophilised preparation was also more oxidized than plasma preparations but not to the same extent as the frozen preparation. This result implies that removal of fibrinogen from plasma, an environment with a high redox buffering capacity, can result in air oxidation of the protein. Evidence for the redox stability of fibrinogen in plasma is provided by the disulfide state analysis shown in **Fig. 1** versus new **Fig. 4**. The plasma used in the experiment in **Fig. 4** was incubated for 16 h at 22°C before the fibrinogen was collected, whereas fibrinogen was collected from plasma immediately after drawing blood in the experiment shown in **Fig. 1**. There is no appreciable difference in the redox state of the fibrinogen disulfides in both experiments.

The following efforts were made to minimise changes in redox state during sample processing. Free cysteines were blocked with ^{12}C -IPA in the native protein and then in the SDS-denatured protein in Laemmli buffer. The concentration (5 mM) and time of incubation with ^{12}C -IPA (60 min) was maximised to achieve complete labelling. Our mass spectrometry data was routinely searched for peptides containing free Cys thiols and these were not detected, which indicates that alkylation of unpaired Cys residues by ^{12}C -IPA or ^{13}C -IPA was complete in the protein. See page 8, paragraph 1 of the revised manuscript. Regarding analysis of fibrin, the polymer was generated in plasma and immediately quenched with 5 mM ^{12}C -IPA to freeze free thiols. The fibrin polymer was then dissociated in acetic acid, which would protonate any cysteine thiols that may not have been blocked with ^{12}C -IPA and prevent possible thiol/disulfide exchange events. We made every effort to eliminate processing artefacts in our experiments and there are no indications in our data of experimental artefacts. Nevertheless, if there was a small degree of disulfide oxidation during processing this would not change the conclusions of our study.

Please discuss the difference in disulfide reduction in fibrinogen from sheared normal plasma compared to the fibrinogen from patients with cardiac failure. What is the suggested mechanism of the observed reduction of disulfide bonds in the circulation?

This cardiac failure patient data has been omitted from the revised manuscript due to the small numbers. We have instead compared the effects of *in vitro* pathological fluid shear on fibrinogen covalent states with a clinical situation associated with high blood shear. This situation is treatment of refractory heart failure patients with extracorporeal membrane oxygenation (ECMO) support. The ECMO device subjects blood to high shear and both thrombosis and bleeding can occur with this intervention. Eight of the 13 fibrinogen disulfide bonds were significantly ($p < 0.001$) more reduced in patients receiving ECMO ($n = 8$) than in healthy human donors ($n = 10$) (see new **Fig. 6B**). These differences are notable considering the variability in disease severity and therapeutic intervention in these patients and suggest that reduction of the fibrinogen disulfides is a result of the blood shear associated with the device. See page 5, paragraph 2 of the revised manuscript. The mechanism of reduction of the disulfides is a key question and one that we are now investigating.

The statement: "To test whether the multiple disulfide-bonded states of fibrinogen is a property of other circulating proteins, we examined von Willebrand factor (VWF)" is misleading since the authors and other groups have already previously shown varying disulfide bond states and disulfide switches under shear conditions for VWF, just in other domains of VWF. In one publication, the VWF redox state was also investigated in patients with heart failure. These publications should be cited and discussed in the context of this manuscript.

Our finding of multiple disulfide-bonded states of blood proteins, including VWF, is distinct from earlier work on blood proteins containing allosteric disulfide bonds. Allosteric disulfides are a subset

of disulfide bonds that control mature protein function when cleaved or formed. These bonds, which are typically just one of the disulfide bonds in a protein (such as the VWF A2 domain disulfide), are nearly always on the protein's surface and accessible to oxidoreductases that control their redox state. In contrast, 12 of the 13 fibrinogen disulfide bonds, with exception of the E domain $\alpha 47$ - $\alpha 47$ disulfide, are mostly hidden from solvent in the interior of the structure (see new **Fig. S6**). This indicates that most of the bonds in the mature protein will be inaccessible to the oxidoreductases and implies their redox state is set during folding and maturation. This has been discussed on page 6, paragraph 2 of the revised manuscript.

REVIEWER COMMENTS

Reviewer #2 (Remarks to the Author):

The authors addressed most comments, however, they also had to change large parts of the manuscripts, where in line with the comments from the reviewers, claims were made without sufficient experimental proof. In general this leaves a slight aftertaste. Also, changes in the manuscript are not labeled, which makes it hard to assess in detail where the manuscript had been changed. Compared to the original submission, this revision is substantially improved, but there are still many points that need to be addressed and discussed.

Major points:

1) The most important major point remains that Cys residues were not blocked at the beginning of the experiment. In most studies, cells are lysed in the presence of an alkylating agent to avoid oxidation of Cys residues. The authors would need to show at least in a limited set of samples that their initial sample processing steps do not affect the oxidation state of Cys residues.

Along this line, the authors added an experiment where they stored two preparations of purified fibrin differently, either frozen at -20C or lyophilised, and based on their data conclude that “removal of fibrinogen from plasma, an environment with a high redox buffering capacity, can result in air oxidation of the protein”. This is a very interesting observation and statement which needs to be further discussed in the light of the (previously in this manuscript) reported observed changes of increased disulfide bridging of plasma proteins – *after* these have been removed from plasma. How reproducible is this “air-oxidation”, what are the ratios obtained for the disulfides and how stable are these?

2) The authors mention that “that fibrinogen 49 in hundreds of different disulfide-bonded covalent states in the circulation that are important for the function of the protein”. Although they have some interesting data on this, it is still too speculative to claim 100s of states are important for the function of the protein. It may be a steady-state which is important to be kept in plasma, which may explain why these disulfide stoichiometries the authors observe have almost no inter-individual variation.

The authors separated for instance fibrinogen after blocking of free Cys on an SDS PAGE gel, with hundreds of disulfide-states I would expect either many bands or largely widened and blurry bands as these disulfides should affect the migration of proteins in a non-denaturing gel.

3) That disulfide forming is conditional does not sound surprising to this reviewer, and I wonder whether this has never been shown before, disulfide formation is part of the protein folding process and depends on energy levels. That there is PTM crosstalk has been long established and also the concept of the ‘PTM code’ is well-known, though not mentioned in this manuscript. Hence, it has

been shown for several proteins that only the whole pattern of PTMs defines the actual protein function.

4) Page 5, line 222: The disulfides are not only prevented from forming, but add a molecule of 133 Da which is more than Averagine, is added to all free Cys residues which can have a major effect on their structure, as it also contains a hydrophobic aromatic structure. So it is possible that the observation is not solely connected to the prevention of disulfide-formation. This is in line with the search parameters used on page 8, line 362, that even consider di- and tri-oxidised Cys residues, that would even more affect the proper formation of fibrin polymers.

Minor comments:

5) Page 2 Line 77: Please provide a number, an average RSD etc. instead of just giving an arbitrary statement such as "little variation". Looking at the figure 1D the variation is indeed remarkably low, probably the lowest I have seen in >10 years of doing proteome analysis on human samples. This definitely needs more discussion.

6) Page 2, line 104: This sentence still needs to be rephrased "The reduced species for each individual disulfide was summed and this number expressed as a percentage of the total". You can sum the *number* of reduced species, or the count. Also, species is plural, so they *were* summed.

7) Fig 4b: The legend says it was one healthy donor, this should be corrected. Green/red combinations should be avoided.

8) Figures 4b and 3b belong together but are from 2 independent experiments. It would be much better to have one figure showing results from one set of experiments, so the same samples from the same preparation.

9) It is kind of counter-intuitive to have the y-axis in the plots referring to the % of reduced disulfides, while basically talking about disulfide formation. I guess one reason the authors chose this representation is that differences look more impressive when showing a reduction of reduced disulfides from 30% to 20% than the other way around, an increase in disulfide formation from 70% to 80%. Nevertheless, it does not help with reading the manuscript. Also, for figure 3b it looks as if some disulfides have a very similar fold-change in reduction of reduced disulfides, most of them seem to be around 30% lower in Fibrin. Please provide the actual values and error margins.

10) Page 2, line 93: 16,384 possibilities is 2^{14} not 2^{13} , so it should be corrected to 8192 possibilities.

11) Page 2, line 77: what is "little"? In a scientific article data should be provided in the text and not only be almost hidden in the figures.

12) Page 8, line 344: This search is important but does not confirm that the first step of blocking was complete, only that the combination of both steps was. This should be a QC step incorporated in the workflow.

13) Page 358: It is higher energy collisional induced dissociation in a QE, not CID

14) Page 361: A mass tolerance of 0.6 Da on MS/MS level is actually unacceptable for a QE, the standard is 0.02 Da or even 0.01 Da. Why did the authors use such an incredibly high mass tolerance? An FDR of 0.01, moreover, sounds unrealistic with such a low complexity sample which probably shouldn't provide the required resolution to actually threshold the data at 1%.

Reviewer #3 (Remarks to the Author):

The authors have investigated the redox state of disulfide bonds in the plasma proteins fibrinogen and α 2-Macroglobulin as well as the changes in the bond states of fibrinogen in different shear environments. They further examined the influence of the treatment of refractory heart failure patients with extracorporeal membrane oxygenation (ECMO) support. ECMO exposes blood to high shear rates, and the author show that this treatment can lead to a reduction of disulfide bonds in fibrinogen. Furthermore, they studied the changes in bond dates during fibrin formation.

Compared to the first version of the manuscript, the authors have significantly improved the manuscript by altering or deleting the concerning data.

The newly added data on changes in disulfide states during fibrin formation and the changes in the fibrinogen bond states due to shear stress are very interesting.

Nonetheless, the conclusions are still exaggerated and do not fully reflect the findings and there still are some concerns which need to be addressed.

1) The authors' statement that fibrinogen exists in hundreds or possibly thousands of different disulfide bonded states is still not supported by the data. The study rather seems to imply the opposite, namely that fibrinogen exists in a restricted number of different states. Although the data show that disulfide bonding is not the same in every fibrinogen molecule and multiple states exist, the authors themselves state that bond formation is conditional. Thus, the data convincingly show that not all possible disulfide-bonded states are present in plasma. This is supported by the data showing that % of reduced disulfides is not 50% for all disulfides. Further, the very similar data from different donors, available crystal structures and the similarity of the bonds of the one investigated donor to those in the crystal structures rather imply that specific states are formed preferentially. Therefore, the conclusions and the discussion should carefully be adjusted to reflect the robust findings of the study.

2) I cannot follow the train of thought in the simulation data for condition 2. How was % reduced disulfides calculated for disulfide 2? When counting, 2 is absent in 12 of the 24 states. Thus, it should be 50%, not something around 30% . Further, disulfides 1, 3 and 4 are also present in 12 of the 24 states but the bar is lower than 50%. If 50% should be correct for 2, this pattern does not resemble

the experimental data. What other states have been simulated? Is there a combination which does resemble the experimental data? Even if 30% was correct for 2, the values for the % reduced disulfides are quite different from those in Fig. 2A.

3) Figure 3: What exactly were the additionally performed experiments? The data points seem to be the same in the revised version for most columns. How come that the difference is now statistically significant for all columns even for those bars for which the data points have not changed, for example C47-C47 and C55-C95?

4) Figure S4B: Is the correct gel shown? The gel bands look the same as for the fibrinogen preparation in Fig. 1B and the same band is indicated for fibrinogen and alpha2M in figures 1B and S4, respectively. Since the proteins were precipitated with specific antibodies and do not have the same molecular weight the gels should be different.

Reviewer #2 (Remarks to the Author):

Major points:

1) The most important major point remains that Cys residues were not blocked at the beginning of the experiment. In most studies, cells are lysed in the presence of an alkylating agent to avoid oxidation of Cys residues. The authors would need to show at least in a limited set of samples that their initial sample processing steps do not affect the oxidation state of Cys residues.

We agree that, ideally, we would collect blood into the alkylating agent, ^{12}C -IPA, and then isolate fibrinogen from this medium. We attempted this in two ways; collection of blood directly into ^{12}C -IPA or addition of ^{12}C -IPA to the collected blood. To achieve a final concentration of 5 mM ^{12}C -IPA in blood we added a 1/20 dilution of stock 100 mM ^{12}C -IPA in DMSO, which is near the solubility limit for ^{12}C -IPA. The solubility limit of ^{12}C -IPA in water is 0.63 mM. Both means of addition of ^{12}C -IPA resulted in lysis of red blood cells (haemolysis) due to the 5% DMSO in the final solution. This is in accordance with known adverse effects of <1% DMSO on red blood cell fragility (Yi et al. FEBS Open Bio 7, 485, 2017; doi.org/10.1002/2211-5463.12193). This information has been included in the re-revised manuscript (page 2, paragraph 3). We had no option, therefore, but to alkylate the fibrinogen in plasma following removal of blood cells. As indicated in our first response to reviews (Reviewer 2, Point 3), the redox state of fibrinogen is stable in plasma for at least 16 h. We have performed an alternative experiment that addresses the issue of sample processing and possible effect on the oxidation state of Cys residues.

We have analysed the fibrinogen constitutively secreted by a human hepatocyte cell line. Liver hepatocytes are the main source of plasma fibrinogen. Serum-free conditioned medium of the hepatocyte cells was collected for 18 h under either standard normoxic (18.8% O_2) or hypoxic (1% O_2) conditions. Hypoxic conditions were employed to test for possible O_2 -mediated oxidation of disulfide bonds in the secreted fibrinogen. The fibrinogen in the conditioned medium was collected on antibody-coated magnetic beads and analysed as for the plasma protein (see **Fig. 1A**). The disulfide bond status of the cell-derived fibrinogen is indistinguishable from plasma fibrinogen produced under either normoxic or hypoxic conditions (new **Fig. S2**). This result provides further evidence that the initial sample processing of the human blood has not affected the oxidation state of the fibrinogen disulfides. The finding also indicates that fibrinogen arrives in plasma in defined covalent states and is not a consequence of post-secretion redox events in the blood. See page 3, paragraph 1 of the revised manuscript.

Along this line, the authors added an experiment where they stored two preparations of purified fibrin differently, either frozen at -20C or lyophilised, and based on their data conclude that “removal of fibrinogen from plasma, an environment with a high redox buffering capacity, can result in air oxidation of the protein”. This is a very interesting observation and statement which needs to be further discussed in the light of the (previously in this manuscript) reported observed changes of increased disulfide bridging of plasma proteins – *after* these have been removed from plasma. How reproducible is this “air-oxidation”, what are the ratios obtained for the disulfides and how stable are these?

We repeated analysis of the purified lyophilised fibrinogen on two more occasions and have included the new data in revised **Fig. S4** (page 34). All disulfides are significantly more oxidized ($p < 0.01$) in purified fibrinogen ($n=4$) than in healthy donor fibrinogen ($n=10$). We have qualified our conclusion by stating that this result suggests (not implies) that removal of fibrinogen from plasma, an environment with a high redox buffering capacity, can result in air oxidation of the protein (page 4, paragraph 1).

The comment ‘increased disulfide bridging of plasma proteins – *after* these have been removed from plasma’ we presume relates to the finding of disulfide oxidation in fibrin polymer.

Fibrin polymer cysteines were quenched with ^{12}C -IPA before removing from the plasma in which the polymer was generated. The redox states of both fibrinogen and fibrin polymer, therefore, were frozen before the proteins were removed from their plasma environment. We have included additional data in the revised manuscript to highlight the change in disulfide status of fibrinogen versus fibrin polymer. In the results presented in **Fig. 3A**, a limiting concentration of thrombin was added to healthy donor plasma, the fibrin polymer removed, another aliquot of thrombin added to the depleted plasma and the new fibrin polymer removed. Fibrinogen concentration in the plasma was depleted 41% in the first reaction and >95% in the second reaction. As we reported in the first revision, there was no change in the disulfide-bonded states of fibrinogen as fibrin formed (**Fig. 3A**). We now include the disulfide-bonded status of the fibrin polymer generated in this experiment (new **Fig. 3A**), which is clearly more oxidised than the fibrinogen from which it derived. See page 15.

2) The authors mention that “that fibrinogen 49 in hundreds of different disulfide-bonded covalent states in the circulation that are important for the function of the protein”. Although they have some interesting data on this, it is still too speculative to claim 100s of states are important for the function of the protein. It may be a steady-state which is important to be kept in plasma, which may explain why these disulfide stoichiometries the authors observe have almost no inter-individual variation. The authors separated for instance fibrinogen after blocking of free Cys on an SDS PAGE gel, with hundreds of disulfide-states I would expect either many bands or largely widened and blurry bands as these disulfides should affect the migration of proteins in a non-denaturing gel

3) That disulfide forming is conditional does not sound surprising to this reviewer, and I wonder whether this has never been shown before, disulfide formation is part of the protein folding process and depends on energy levels. That there is PTM crosstalk has been long established and also the concept of the ‘PTM code’ is well-known, though not mentioned in this manuscript. Hence, it has been shown for several proteins that only the whole pattern of PTMs defines the actual protein function.

These are fair points and we have modified our discussion of the number of possible disulfide-bonded states of fibrinogen in the re-revised manuscript. Our intention was to provide a range for the possible number of disulfide-bonded states but appreciate that this discussion is perhaps too speculative with the information currently at hand. Reviewer 3 has raised similar concerns. We have, therefore, amended the discussion of this point to the following (page 3, paragraph 2).

‘These results indicate that circulating fibrinogen exists in multiple disulfide-bonded states. A protein containing n disulfide bonds, where the bonds are either formed or broken, has 2^n possible disulfide-bonded states. This situation is illustrated in a model polypeptide containing 5 disulfide bonds that can exist in 32 (2^5) possible disulfide-bonded states (**Fig. S3A**). The different states are represented in cartoon form in **Fig. S3B**. In the case of the 13 fibrinogen disulfides that were measured, this analysis equates to a maximum 8,192 possible disulfide-bonded states of the protein.’

Original **Fig. 3** has been modified and is now **Fig. S3**. The α 2-macroglobulin result has been moved from **Fig. S4** to **Fig. 6**.

We have resolved plasma fibrinogen on a non-denaturing gel (NuPAGE Novex Tris Acetate 3-8% gel) and it runs as a blurred band (Coomassie stained protein shown at left). We are currently examining whether this smear is a consequence of the multiple disulfide-bonded forms of the protein.

4) Page 5, line 222: The disulfides are not only prevented from forming, but add a molecule of 133 Da which is more than Averagine, is added to all free Cys residues which can have a major effect on their structure, as it also contains a hydrophobic aromatic structure. So it is possible that the observation is not solely connected to the prevention of disulfide-formation. This is in line with the search parameters used on page 8, line 362, that even consider di- and tri-oxidised Cys residues, that would even more affect the proper formation of fibrin polymers.

This is a good point. We have added the following statement to the re-revised manuscript: 'We cannot exclude the possibility, however, that IPA alkylation of one or more cysteines results in steric effects on fibrin polymer formation.' See page 4, paragraph 2 of the re-revised manuscript.

Minor comments:

5) Page 2 Line 77: Please provide a number, an average RSD etc. instead of just giving an arbitrary statement such as "little variation". Looking at the figure 1D the variation is indeed remarkably low, probably the lowest I have seen in >10 years of doing proteome analysis on human samples. This definitely needs more discussion.

We agree that the variation between healthy human donors is remarkably low and is also the lowest we have seen in our studies of human samples. We have included coefficient of variation calculations in the re-revised manuscript. See page 2, paragraph 4 of the re-revised manuscript. In the Discussion, we speculate that the balance of the forms relates to thermodynamics considerations during protein maturation. If so, this could account for the small inter-individual variation. See page 6, paragraph 1.

6) Page 2, line 104: This sentence still needs to be rephrased "The reduced species for each individual disulfide was summed and this number expressed as a percentage of the total". You can sum the *number* of reduced species, or the count. Also, species is plural, so they *were* summed.

This analysis has been removed from the re-revised manuscript (see point 2).

7) Fig 4b: The legend says it was one healthy donor, this should be corrected. Green/red combinations should be avoided.

Both these corrections have been made in the re-revised manuscript. See page 15.

8) Figures 4b and 3b belong together but are from 2 independent experiments. It would be much better to have one figure showing results from one set of experiments, so the same samples from the same preparation.

The results shown in **Fig. 4B** logically follow from the results in **Fig. 4A**. We feel they are best understood in this context.

9) It is kind of counter-intuitive to have the y-axis in the plots referring to the % of reduced disulfides, while basically talking about disulfide formation. I guess one reason the authors chose this representation is that differences look more impressive when showing a reduction of reduced disulfides from 30% to 20% than the other way around, an increase in disulfide formation from 70% to 80%. Nevertheless, it does not help with reading the manuscript. Also, for figure 3b it looks as if some disulfides have a very similar fold-change in reduction of reduced disulfides, most of them seem to be around 30% lower in Fibrin. Please provide the actual values and error margins.

We could have presented the data either way but feel it would make no difference. The expectation was that that all disulfides are 100% oxidised. Presenting the data as % of bond that is

cleaved, rather than % of bond that is oxidised, follows logically in our opinion. The values and error margins are shown in new **Table S4** (page 25).

10) Page 2, line 93: 16,384 possibilities is 2^{14} not 2^{13} , so it should be corrected to 8192 possibilities.

We have corrected this error in the re-revised manuscript (page 3, paragraph 2).

11) Page 2, line 77: what is “little”? In a scientific article data should be provided in the text and not only be almost hidden in the figures.

This has been amended in the re-revised manuscript. The section now reads: ‘The bonds ranged from 10 to 50% reduced and there was remarkably small donor-to-donor variation and no significant gender difference. The coefficients of variation ranged from a low of 3.9% for the β C424- β C437 disulfide to a high of 16.5% for the α C68- β C106 bond.’ See page 2, paragraph 4.

12) Page 8, line 344: This search is important but does not confirm that the first step of blocking was complete, only that the combination of both steps was. This should be a QC step incorporated in the workflow.

This QC step is now part of our workflow.

13) Page 358: It is higher energy collisional induced dissociation in a QE, not CID

We have corrected this error in the re-revised manuscript.

14) Page 361: A mass tolerance of 0.6 Da on MS/MS level is actually unacceptable for a QE, the standard is 0.02 Da or even 0.01 Da. Why did the authors use such an incredibly high mass tolerance? An FDR of 0.01, moreover, sounds unrealistic with such a low complexity sample which probably shouldn’t provide the required resolution to actually threshold the data at 1%.

The mass tolerance for the QE plus used for MS and MS/MS data acquisition was set at 0.02 Da. The stated parameters were for search of disulfide-linked peptides using Byonic analysis software (Protein Metrics). Since we defined the mass tolerance for MS as 0.01 Da (or 10 ppm), the total mass error for MS/MS cannot exceed that of 0.01 Da even when we set the MS/MS at 0.6 Da. When we manually curated our data using Xcalibur we searched for MS and MS/MS at 0.006 Da (or 6 ppm). The lowest FDR in Byonic is 1%. The p-values for the disulfide bonds we identified were all <0.01 and these values have been included in the re-revised manuscript (**Table S3**) (page 25).

Reviewer #3 (Remarks to the Author):

1) The authors’ statement that fibrinogen exists in hundreds or possibly thousands of different disulfide bonded states is still not supported by the data. The study rather seems to imply the opposite, namely that fibrinogen exists in a restricted number of different states. Although the data show that disulfide bonding is not the same in every fibrinogen molecule and multiple states exist, the authors themselves state that bond formation is conditional. Thus, the data convincingly show that not all possible disulfide-bonded states are present in plasma. This is supported by the data showing that % of reduced disulfides is not 50% for all disulfides. Further, the very similar data from different donors, available crystal structures and the similarity of the bonds of the one investigated donor to those in the crystal structures rather imply that specific states are formed preferentially. Therefore, the conclusions and the discussion should carefully be adjusted to reflect the robust findings of the study.

These are fair points and we have modified our discussion of the number of possible disulfide-bonded states of fibrinogen in the re-revised manuscript. Our intention was to provide a range for the possible number of disulfide-bonded states but appreciate that this discussion is perhaps too speculative with the information currently at hand. Reviewer 2 has raised similar concerns. We have, therefore, amended the discussion of this point to the following (page 3, paragraph 2).

‘These results indicate that circulating fibrinogen exists in multiple disulfide-bonded states. A protein containing n disulfide bonds, where the bonds are either formed or broken, has 2^n possible disulfide-bonded states. This situation is illustrated in a model polypeptide containing 5 disulfide bonds that can exist in 32 (2^5) possible disulfide-bonded states (**Fig. S3A**). The different states are represented in cartoon form in **Fig. S3B**. In the case of the 13 fibrinogen disulfides that were measured, this analysis equates to a maximum 8,192 possible disulfide-bonded states of the protein.’

Original **Fig. 3** has been modified and is now **Fig. S3**. The α 2-macroglobulin result has been moved from **Fig. S4** to **Fig. 6**.

2) I cannot follow the train of thought in the simulation data for condition 2. How was % reduced disulfides calculated for disulfide 2? When counting, 2 is absent in 12 of the 24 states. Thus, it should be 50%, not something around 30%. Further, disulfides 1, 3 and 4 are also present in 12 of the 24 states but the bar is lower than 50%. If 50% should be correct for 2, this pattern does not resemble the experimental data. What other states have been simulated? Is there a combination which does resemble the experimental data? Even if 30% was correct for 2, the values for the % reduced disulfides are quite different from those in Fig. 2A.

We apologise for this embarrassing error. We mistakenly included in **Fig. 2D(2)** the situation where disulfide 5 forms only if disulfide 4 has formed - not 5 forms only if 2 has formed. The corrected figure follows.

In response to Point 1 (and similar concerns from Reviewer 2) this analysis has been omitted from the re-revised manuscript.

3) Figure 3: What exactly were the additionally performed experiments? The data points seem to be the same in the revised version for most columns. How come that the difference is now statistically significant for all columns even for those bars for which the data points have not changed, for example C47-C47 and C55-C95?

We apologise for this confusion. The data in **Fig. 3** has not changed from the original manuscript, only the description of the data. We originally described that certain fibrin disulfides were more oxidised than others. In response to Reviewer 1, we now simply present the p values for each disulfide. The values and error margins for the data in **Fig. 3** are shown in new **Table S4** (page 25).

4) Figure S4B: Is the correct gel shown? The gel bands look the same as for the fibrinogen preparation in Fig. 1B and the same band is indicated for fibrinogen and alpha2M in figures 1B and S4, respectively. Since the proteins were precipitated with specific antibodies and do not have the same molecular weight the gels should be different.

The correct gels are shown. Fibrinogen has a molecular mass of 340 kDa, while α 2-macroglobulin is a homo-tetramer with a molecular mass of 720 kDa. Both proteins resolve above the highest Mr standard at the top of the gel. These masses have been included in the figure legends in the re-revised manuscript.

REVIEWERS' COMMENTS

Reviewer #2 (Remarks to the Author):

I appreciate the authors' efforts to address most comments, but also want to note that several central statements of the first and also second submission turned out to be wrong or at least too speculative.

The manuscript clearly improved considerably during this reviewing process.

Reviewer #3 (Remarks to the Author):

Response to point 1: although the authors have adjusted their conclusions in the results part the discussion has barely been changed. Also the very good and concerning comment of reviewer 2 that the addition of IPA could alter fibrin structure is not mentioned in the discussion. Thus, the discussion does still not fully reflect the shortcomings of the study.

Response to point 2: the figure is still incorrect. Firstly it should be 24 states not 23 as state 27 could still be formed. Also percentage for 1, 3 and 4 should be 52% when calculated with 23 states. It seems that here the value for the oxidized form of 48% is shown. But it does not really matter as the figure now also has been omitted from the manuscript.

The other two comments have been addressed.

Line 68: Just out of curiosity: Why does haemolysis disturb the experiment?

Minor comment: Fig. S4 in the graph legend, the grey dot should be red.

Reviewer #3 (Remarks to the Author):

Response to point 1: although the authors have adjusted their conclusions in the results part the discussion has barely been changed. Also the very good and concerning comment of reviewer 2 that the addition of IPA could alter fibrin structure is not mentioned in the discussion. Thus, the discussion does still not fully reflect the shortcomings of the study.

The discussion has been revised and expanded as indicated. We have highlighted the potential shortcoming of the IPA blocking experiment, and have also expanded discussion of the fluid shear findings. See pages 5 and 6.

Response to point 2: the figure is still incorrect. Firstly it should be 24 states not 23 as state 27 could still be formed. Also percentage for 1, 3 and 4 should be 52% when calculated with 23 states. It seems that here the value for the oxidized form of 48% is shown. But it does not really matter as the figure now also has been omitted from the manuscript.

The reviewer is correct (again), although this is now not an issue.

Line 68: Just out of curiosity: Why does haemolysis disturb the experiment?

Lysed red blood cells could release factors into the plasma that influence the disulfide status of fibrinogen.

Minor comment: Fig. S4 in the graph legend, the grey dot should be red.

This has been corrected in the revised manuscript.